# High-affinity tuning of single fluorescent protein-type indicators by flexible linker length optimization in topology mutant
Yusuke Hara [1], Aya Ichiraku[1], Tomoki Matsuda [2], Ayuko Sakane [3,4], Takuya Sasaki [3], Takeharu Nagai [2,5] & Kazuki Horikawa [1] ✉

Genetically encoded $Ca^{2+}$ indicators (GECIs) are versatile for live imaging of cellular activities. Besides the brightness and dynamic range of signal change of GECIs, $Ca^{2+}$ affinity is another critical parameter for successful $Ca^{2+}$ imaging, as the concentration range of $Ca^{2+}$ dynamics differs from low nanomolar to sub-millimolar depending on the celltype and organism. However, ultrahigh-affinity GECIs, particularly the single fluorescent protein (1FP)-type, are lacking. Here, we report a simple strategy that increases $Ca^{2+}$ affinity through the linker length optimization in topology mutants of existing 1FP-type GECIs. The resulting ultrahigh-affinity GECIs, CaMPARI-nano, BGECO-nano, and RCaMP-nano ($K_d$ = 17–25 nM), enable unique biological applications, including the detection of low nanomolar $Ca^{2+}$ dynamics, highlighting active signaling cells, and multi-functional imaging with other second messengers. The linker length optimization in topology mutants could be applied to other 1FP-type indicators of glutamate and potassium, rendering it a widely applicable technique for modulating indicator affinity.

The intracellular calcium ion ($[Ca^{2+}]_{in}$) is a ubiquitous second messenger that controls physiological and pathological phenomena[1], such as neuronal activity, hormonal sensing, and wound healing. Analyzing its dynamics with fine spatiotemporal resolution is essential to understanding the functions of $[Ca^{2+}]_{in}$. Synthetic dyes or genetically encoded $Ca^{2+}$ indicators (GECIs) are powerful tools for such analyses[2]. GECIs are particularly useful for $Ca^{2+}$ imaging in specific cells and organelles. This is because they can be genetically localized to these targets, which is challenging for synthetic dyes. Generally, GECIs consist of fluorescent or bioluminescent reporters and sensor motifs for $Ca^{2+}$, such as calmodulin (CaM) and the CaM-binding peptide M13 or RS20[2]. There are two types of GECIs: single-fluorescent protein (1FP)-type (e.g., GCaMPs, GECOs) and Förster resonance energy transfer (FRET)-type (e.g., chameleons). These indicators report changes in $[Ca^{2+}]_{in}$ intensiometrically or ratiometrically by modulating the optical properties of fluorescent reporters through $Ca^{2+}$-dependent interactions with the sensor motifs.

To monitor changes in $[Ca^{2+}]_{in}$ using GECIs, it is essential to carefully select indicators that have an appropriate affinity for $Ca^{2+}$ within the range of changes in target cells and organelles. This is because resting $[Ca^{2+}]_{in}$ and the amplitude of $[Ca^{2+}]_{in}$ transients can vary greatly, ranging from low nanomolar to sub-millimolar[1]. Current GECIs are optimized for relatively high $[Ca^{2+}]_{in}$ concentrations above 100 nM[2]. However, they are not suitable for detecting $[Ca^{2+}]_{in}$ dynamics in certain living samples with low nanomolar levels (<100 nM) of resting $[Ca^{2+}]_{in}$ and amplitude of $[Ca^{2+}]_{in}$ transients, such as those found in malaria parasites[3], plant cells[4], murine astroglia[5], and slim molds[6] (Supplementary Fig. 1). To monitor $[Ca^{2+}]_{in}$ dynamics at low nanomolar levels, ultrahigh-affinity indicators whose $K_d$ is in the middle of the $[Ca^{2+}]_{in}$ ranges are required (Supplementary Fig. 1). Yellow Chameleon-Nano (YC-Nano), a FRET-type GECI, is an exceptional indicator with ultrahigh-affinity[6]. Although YC-Nano with $K_d$s of 15–65 nM enables visualization of $[Ca^{2+}]_{in}$ dynamics in the low nanomolar range, its applications are occasionally hindered by technical limitations inherent to FRET-type indicators, such as small signal changes and wide occupancy of the spectral range. However, these limitations can be overcome by using 1FP-type GECIs with ultrahigh-affinity, which have not been available until now.

[1]Department of Optical Imaging, Advanced Research Promotion Center, Tokushima University, 3-18-15 Kuramoto, Tokushima, Tokushima 770-8503, Japan. [2]Department of Biomolecular Science and Engineering, SANKEN, Osaka University, Mihogaoka 8-1, Ibaraki, Osaka 567-0047, Japan. [3]Department of Biochemistry, Tokushima University Graduate School of Medicine, 3-18-15 Kuramoto, Tokushima, Tokushima 770-8503, Japan. [4]Division of Interdisciplinary Researches for Medicine and Photonics, Institute of Post-LED Photonics (pLED), Tokushima University, 3-18-15 Kuramoto, Tokushima, Tokushima 770-8503, Japan. [5]Institute for Open and Transdisciplinary Research Initiatives, Osaka University, Yamadaoka 2-1, Suita, Osaka 565-0871, Japan. ✉e-mail: horikawa.kazuki@tokushima-u.ac.jp

Affinity tuning of GECIs has mainly been performed by modulating sensor motifs to affect the kinetics of $Ca^{2+}$-dependent conformational changes through motif-substitution[7] and mutations. While several effective mutations decreasing $Ca^{2+}$ affinity have been identified[8,9], few mutations increasing affinity have been reported[8,10]. An alternative method for engineering high-affinity indicators has been described for FRET-type GECIs, which involves enhancing sensor motifs interactions. In FRET-type GECIs, the sensor motifs (CaM and M13) are structurally connected by a linker. Increasing the flexibility of this linker reduces steric hindrance, allowing for enhanced $Ca^{2+}$-dependent interactions of sensor motifs and thereby increasing indicator affinity[6]. While this strategy is not applicable to most 1FP-type GECIs whose sensor motifs are physically separated at N- and C-terminal of the core FPs, recently reported topology mutant of GECIs is interesting. The topology mutant is the circularly permuted form of GECIs in which FP was inserted with CaM-M13 having a similar molecular organization with one of the most classical designs of GECIs like camgaroo[11]. Importantly, the topology mutant of GCaMP6 showed increased $Ca^{2+}$ affinity ($K_d = 283$ nM) than the original GCaMP6 ($K_d = 380$ nM) while preserving a large signal change and brightness[12]. However, the optimization of linker length, as reported for YC-Nano, has not been examined in topology mutants of 1FP-type GECIs.

Here, we hypothesized that the linker length optimization in topology mutants of GECIs would facilitate the $Ca^{2+}$-dependent interaction between CaM and RS20, preventing their dissociation and thereby increasing $Ca^{2+}$ affinity. Therefore, we aimed to modify existing 1FP-type GECI designs by circular permutation[12] in combination with linker length optimization[6] and conventional mutagenesis. We demonstrate that the linker length optimization in topology mutant increases the affinity of existing 1FP-type GECIs, such as CaMPARIs[9,10] and GECOs[13], which are useful for $Ca^{2+}$ imaging in the low nanomolar range. The developed indicators can be applied for functional labeling and multi-functional imaging of actively signaling cells displaying transient changes in $[Ca^{2+}]_{in}$.

## Results

### Engineering of ultrahigh-affinity GECIs by the linker length optimization in topology mutant

To develop 1FP-type GECIs with ultrahigh-affinity suitable for detecting $[Ca^{2+}]_{in}$ at the low nM range (15–30 nM, covering one of the lowest resting $[Ca^{2+}]_{in}$ in *Dictyostelium discoideum* cells; Supplementary Fig. 1), we aimed to enhance the $Ca^{2+}$ affinity of CaMPARI2[10]. Among the CaMPARI2 variants, CaMPARI2 F391W—referred to as CaMP2_F391W—showed the highest $Ca^{2+}$ affinity ($K_d = 121$ nM) through mutations in its RS20 moiety. To screen for other mutations that could further increase the $Ca^{2+}$ affinity, we performed site-directed mutagenesis of RS20 and identified 13 mutations at 6 different sites in 106 mutations (Supplementary Fig. 2a). The combination of three of these mutations (G395A, H396W, L398I) resulted in the lowest $K_d$ (Supplementary Fig. 2b), measuring 43 nM (CaMP2_F391W+3mut; Supplementary Fig. 3a-ii, b-ii). However, this value fell short of our target affinity of around 15–30 nM (Supplementary Fig. 1).

We next tried the linker length optimization that worked for YC-nano[6]. To allow linker length optimization, we first generated the topology mutant of CaMP2_F391W by circular permutation with expecting increased $Ca^{2+}$ affinity as reported for GCaMP6[12]. The entire organization of CaMP2_F391W was circularly permuted by connecting the tail of RS20 to the head of CaM, and the circularly permuted mEos2 moiety was split again to restore the original configuration of mEos2 (Fig. 1a; Supplementary Fig. 3a-iii). For a linker connecting RS20 and CaM, we first tested 24 amino acids (a.a.) of flexible one ([GGTGGS]$_4$). The resulting cp-CaMP2_F391W_linker24, in which RS20-CaM was inserted at position 143/144 of mEos2, showed increased $Ca^{2+}$ affinity ($K_d = 54$ nM; Supplementary Table 1) compared to the parental CaMP2_F391W. A slight increase of $Ca^{2+}$ affinity was observed for cp-CaMP2_F391W_linker12, which incorporates a 12 a.a. linker ([GGTGGS]$_2$) ($K_d = 46$ nM; Supplementary Fig. 3a-iii, b-iii, c), while it was not for a 6 a.a. linker ([GGTGGS]$_1$)

($K_d = 111$ nM; Supplementary Fig. 3c, Supplementary Table 1). These results demonstrate that the topology mutant of CaMP2_F391W shows higher $Ca^{2+}$ affinity as reported for GCaMP6[12], and there is an optimal length of flexible linker for a high $Ca^{2+}$ affinity as reported for YC-nano[6].

To achieve higher affinity, we tested the combined effects of the linker length optimization in topology mutants and mutations in RS20. We introduced three high-affinity mutations in RS20 (G395A, H396W, L398I) of cp-CaMP2_F391W_linker12, resulting in cp-CaMP2_F391W_linker12+3mut. This variant exhibited a $K_d$ value of 19 nM (Fig. 1a bottom, 1b green; Supplementary Fig. 3a-iv, 1b-iv) within the target affinity range (Supplementary Fig. 2). The optical properties of cp-CaMP2_linker12+3mut were comparable to those of the parental CaMP2_F391W (Supplementary Table 1). These properties included brightness (Supplementary Fig. 3i), dynamic range, spectral properties (Supplementary Fig. 3f–h), $pK_a$ (Supplementary Fig. 3j), and photoconvertibility (Fig. 1c; Supplementary Fig. 4). We named the ultrahigh-affinity green indicator, derived from cp-CaMP2_F391W_linker12+3mut, as "CaMPARI-nano."

To validate the effectiveness of topology mutant and its linker length optimization in improving the affinity of GECIs, we employed a similar engineering approach for other GECIs that have slightly different molecular configurations (Supplementary Fig. 5). We selected B-GECO1[13] ($K_d = 154$ nM) and R-CaMP1.01[14] ($K_d = 222$ nM), which are blue and red GECIs as our templates (Supplementary Table 1). These indicators also feature RS20 and CaM, but their connection order to the fluorescent reporter was reversed compared to CaMPARIs. Topology mutants of B-GECO1 and R-CaMP1.01 were generated by employing 3, 6, 12, and 24 a.a. of flexible linkers (Supplementary Fig. 6a, 7a). All the resulting topology variants showed decreased $K_d$ values (Supplementary Fig. 6b, 7b) in which the 6 a.a. linker yielded the highest $Ca^{2+}$ affinity (66 nM and 105 nM for B-GECO1 and R-CaMP1.01, respectively) (Supplementary Fig. 6b, 7b). By further combinations with the high-affinity mutations in RS20 that also increased $Ca^{2+}$ affinity of parental B-GECO1 and R-CaMP1.01 ($K_d = 60$ and 38 nM for B-GECO1+3mut and RCaMP+3mut, respectively), we eventually obtained a blue indicator "BGECO-nano" ($K_d = 17$ nM; Fig. 1d; Supplementary Fig. 6b) and a red indicator "RCaMP-nano" ($K_d = 24$ nM; Fig. 1e; Supplementary Fig. 7b), being in our target affinity range. The biochemical properties of these indicators were also comparable to those of their parental indicators (Supplementary Fig. 6e–h, 7e–h; Supplementary Table 1). The successful development of the three-color ultrahigh-affinity indicators indicates that the linker length optimization in topology mutants is an effective approach to improving $Ca^{2+}$ affinity, in addition to conventional mutagenesis.

A high $Ca^{2+}$ affinity is achieved by accelerated association and/or slowed-down dissociation between $Ca^{2+}$ and sensing motifs. To decipher how the linker length optimization and RS20-mutations contributed in enhancing $Ca^{2+}$ affinity, we conducted a kinetic analysis of developed GECIs (Supplementary Table 1). The results showed different contributions of two strategies on on- and off-kinetics depending on GECI species. In CaMPARIs and BGECOs, the linker length optimization alone affected both kinetics (Supplementary Fig. 3d, 6c), while it specifically accelerated on-kinetics in RCaMPs as observed for YC-nano[6] (Supplementary Fig. 7c). RS20-mutations also differently affected on- and off-kinetics in such a way that off-kinetics was slowed down in CaMPARIs (Supplementary Fig. 3e), on-kinetics was accelerated in RCaMPs (Supplementary Fig. 7d), and the both kinetics were affected in BGECOs (Supplementary Fig. 6d). Combination of two modifications further complicated the kinetic effects (Supplementary Note 1), but ultrahigh-affinity of CaMPARI-nano and BGECO-nano was achieved by both fast association and slow dissociation (Supplementary Fig. 3e, 6d), while that of RCaMP-nano was solely achieve by accelerated on-kinetics with preserving fast decay (Supplementary Fig. 7d). The relationship between $K_d$ and $k_{off}$ also highlighted fast decay kinetics of RCaMP series (Fig. 1f), suggesting a potential suitability of RCaMP-nano in imaging of fast $Ca^{2+}$ dynamics at the low nM range.

**Fig. 1 | Engineering of 1FP-type GECIs with increased Ca²⁺ affinity. a** Engineering of CaMPARI-nano. The whole structure of parental CaMPARI2 (CaMP2_F391W) was circularly permutated to link RS20 and CaM with a long, flexible linker. The numbering of high-affinity mutations in RS20 represents the position in the parental CaMPARI2[10]. **b** Ca²⁺ titration of CaMPARI-nano (green) and the parental CaMP2_F391W (black). **c** Photoconversion time-course of purified CaMPARI-nano (red) and the parental CaMP2_F391W (black) in the presence (solid lines) or absence (dashed lines) of Ca²⁺, as a function of exposure to photoconversion light (405 nm, 250 mW cm⁻²). **d** Ca²⁺ titration of BGECO-nano (blue) and the parental B-GECO1 (black). **e** Ca²⁺ titration of RCaMP-nano (red) and the parental R-CaMP1.01 (black). See Supplementary Figs. 5–7 for the molecular design. **f** $K_d$ and $k_{off}$ for parental and developed GECIs, including jGCaMP8s as a reference.

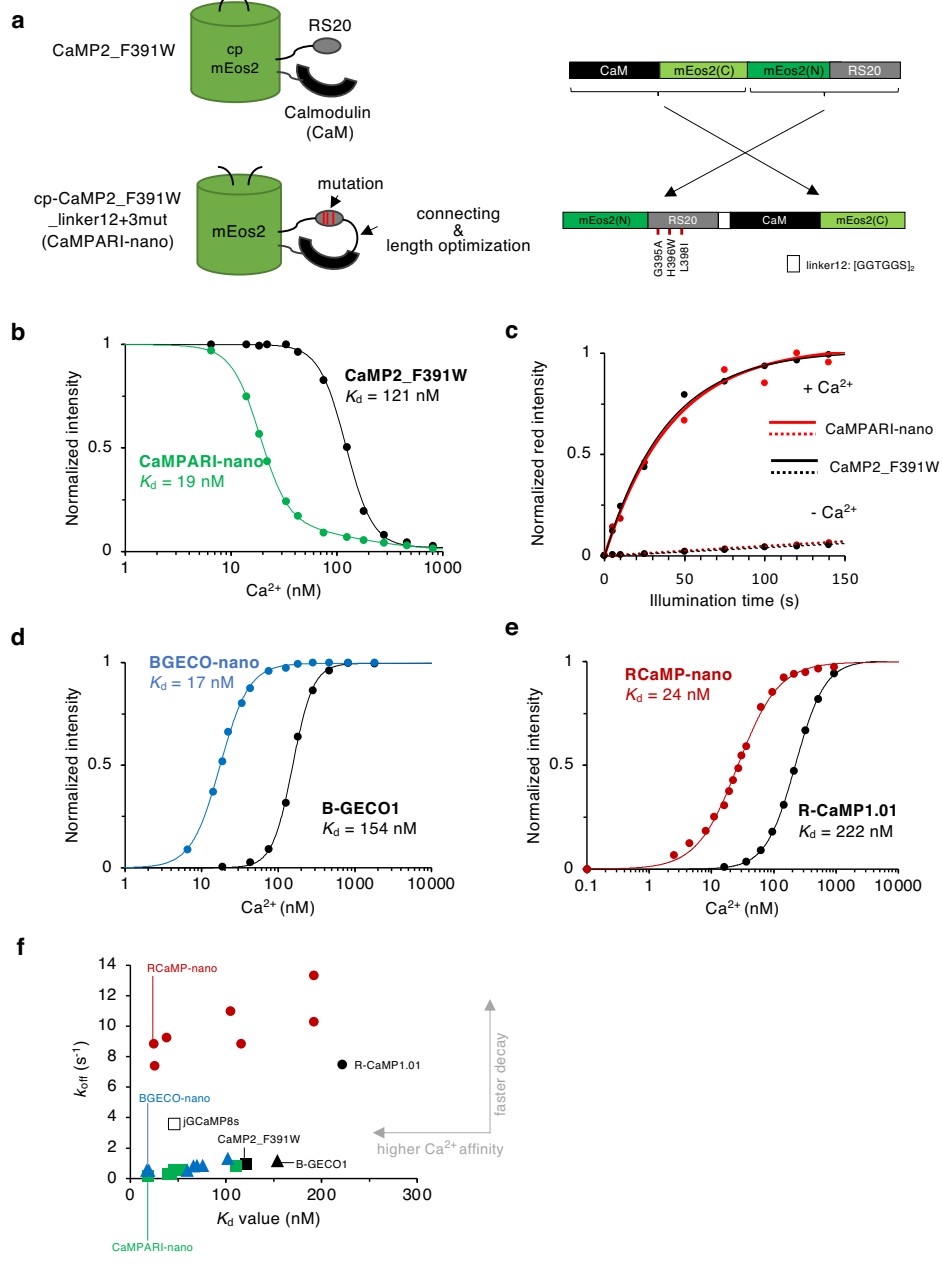

## Detection of [Ca²⁺]ᵢₙ dynamics in the low nanomolar range

To evaluate the capability of high-affinity GECIs to detect physiological [Ca²⁺]ᵢₙ changes in the low nanomolar range, we performed comparative Ca²⁺ imaging using low-affinity (*i.e.*, parental) and high-affinity (*i.e.*, developed) indicators. We utilized *Dityostelium discoideum* cells (*D. discoideum*), which are known to have a low level of resting [Ca²⁺]ᵢₙ (~10 nM). When stimulated with extracellular cyclic adenosine monophosphate ([cAMP]ₑₓ), a chemoattractant molecule, these cells exhibit [Ca²⁺]ᵢₙ transients below 100 nM[6]. We obtained *D. discoideum* cells expressing either CaMPARI-nano or its parental indicator and monitored the changes in fluorescence intensity upon stimulation with high or low concentrations of [cAMP]ₑₓ. The magnitude of the signal change was quantitatively analyzed using the ratio (*R*) of the fluorescence intensity of CaMPARIs to that of the fused reference marker (mRFPmars). This allowed for a quantitative comparison of both the baseline intensity and the signal changes among cells expressing different indicators with varying Ca²⁺ affinities.

The fluorescence intensity of the CaMPARIs decreased with increasing [Ca²⁺]ᵢₙ. At the resting state, CaMPARI-nano showed lower intensity than the parental CaMP2_F391W, which was quantitatively confirmed by ratio analysis (CaMPARI-nano, *R* = 1.0; CaMP2_F391W, *R* = 1.2; Fig. 2a top). Next, we evaluated the ratio changes for [Ca²⁺]ᵢₙ transients. When cells were stimulated with a bath application of a high concentration of [cAMP]ₑₓ (10 μM), which induces a large [Ca²⁺]ᵢₙ transient[6], both indicators showed large ratio changes (*ΔR* of 0.7 for CaMPARI-nano, 0.25 for CaMP2_F391W; Fig. 2a top-left). We also attempted to detect physiological [Ca²⁺]ᵢₙ transients triggered by low concentrations of [cAMP]ₑₓ. The nutrient-starved *D. discoideum* cell population is a suitable model for such a test because these cells spontaneously synthesize and secrete cAMP, which induces a small amplitude of [Ca²⁺]ᵢₙ transients in neighboring cells[6]. In such cell populations, CaMPARI-nano detected the transient increase in [Ca²⁺]ᵢₙ (*ΔR* = 0.5), whereas the parental indicator could not (*ΔR* < 0.05) (Fig. 2a top-right, Fig. 2b; Supplementary Movie 1). To quantitatively evaluate the utility of ultrahigh-affinity GECIs, we compared the magnitude of signal change (Fig. 2c), peak SNR (Fig. 2d), and decay kinetics (Fig. 2e) to that of parental and related construct. Specifically, to enable simultaneous comparison of

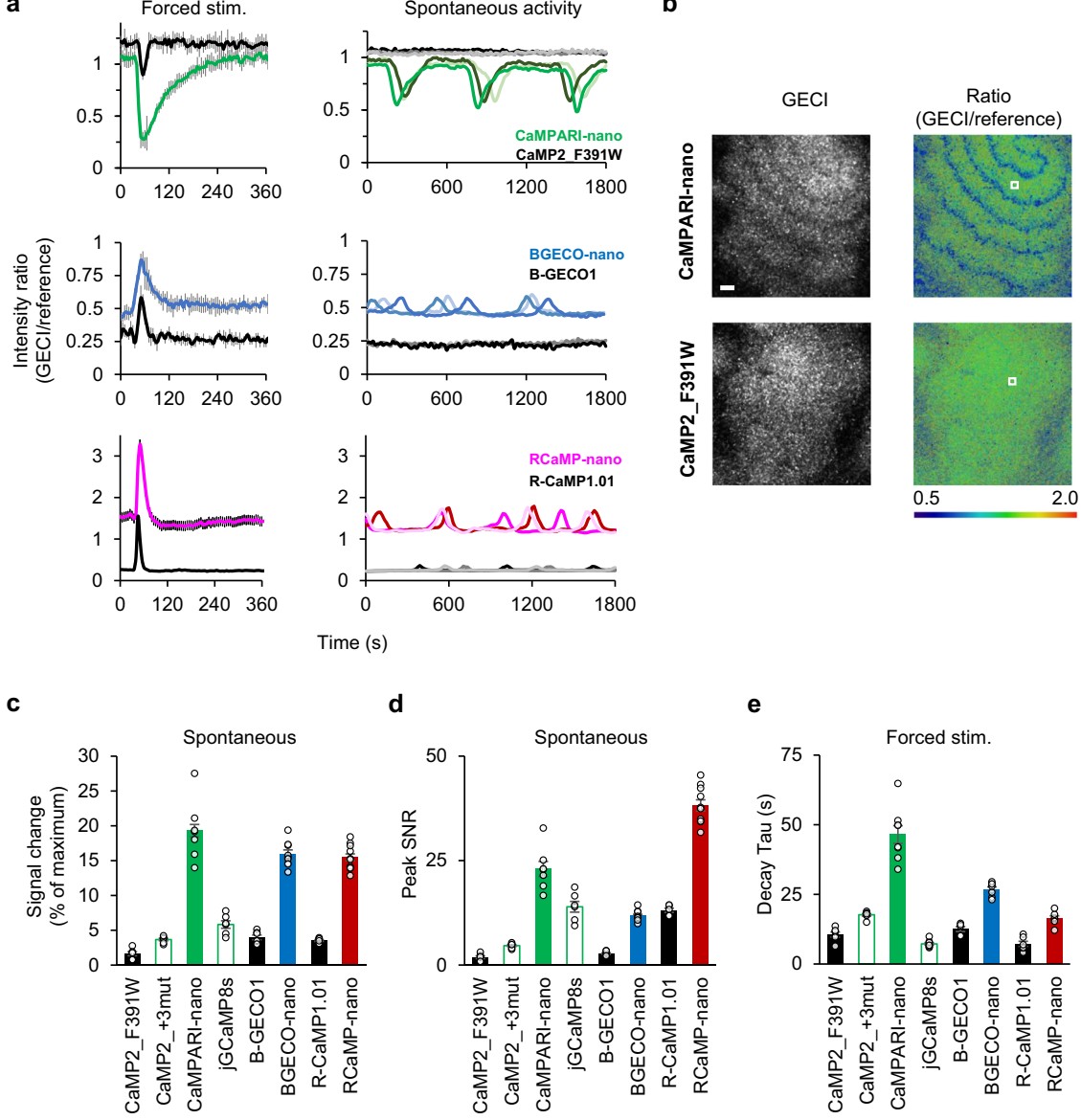

**Fig. 2 | Increased detection sensitivity of spontaneous [Ca²⁺]ᵢₙ transients by ultrahigh-affinity GECIs. a** The measurement of $[Ca^{2+}]_{in}$ dynamics induced by bath application of 10 μM $[cAMP]_{ex}$ at t = 30 s (left) or spontaneously generated $[cAMP]_{ex}$ in the population of chemotactic cells (right). The intensity ratio of GECIs over the fused reference maker was shown for quantitative analysis. Mean ±s.e.m. of 10 cells for left. Population average in ROIs containing ~10 cells was shown for three ROIs for right. **b** Signaling waves of $[Ca^{2+}]_{in}$ in a cell population cultured on a non-nutrient agar plate. Raw GECI's intensity (left), intensity ratio (GECI/reference, middle). Scale bar: 0.1 mm. **c, d** The magnitude of signal change, defined as the ratio of a given response to the maximum signal change in each GECI (**c**), and the peak signal-to-noise-ratio. **c** And the peak signal-to-noise-ratio (SNR, **d**) for spontaneous $[Ca^{2+}]_{in}$. See Supplementary Fig. 8a for the quantification method used in Fig. 2c. Mean ± s.e.m. of 5 measurements. **e** Tau of the signal decay in (**a**). Mean ± s.e.m of 10 cells.

the signal change magnitude between both flashing and inversely flashing GECIs of different colors, we measured the fractional change of given indicators to their maximal response (see Supplementary Fig. 8 for a more detailed definition), since the commonly utilized $\Delta F/F_{resting}$ is not suitable for comparing the performance of flashing and inversely flashing GECIs. We also included recently developed jGCaMP8s as a representative of existing GECIs having high $Ca^{2+}$-affinity and fast kinetics[7], since iNTnC2[15], another high-affinity GECI, was found to show insufficient performance in our in vitro and *in-cell* assay (Supplementary Note 2). The signal change and peak SNR were suboptimal for CaMP2_F391W+3mut or jGCaMP8s having $K_d$ of 40–50 nM (Fig. 2c, d; Supplementary Fig. 8a), demonstrating that $K_d$ = 20–30 nM is advantageous for robust detection physiological $[Ca^{2+}]_{in}$ transients triggered by spontaneously secreted $[cAMP]_{ex}$. The above results were similarly reproduced using BGECO-nano and RCaMP-nano (Fig. 2a middle and bottom, 2c–e; Supplementary

Movie 1) with ignorable photobleaching (Supplementary Fig. 8b). Together with increased detectability, fast and slow decay kinetics of RCaMP-nano and CaMPARI-nano (Fig. 2e) would suggest their suitability for imaging fast $[Ca^{2+}]_{in}$ dynamics and functional highlighting by photo-conversion, respectively.

## Utility of high-affinity GECIs in mammalian cells
To test the applicability of high-affinity GECIs in mammalian cells, we investigated resting $[Ca^{2+}]_{in}$ in several cell lines. Resting $[Ca^{2+}]_{in}$ and its cellular variations is important determinants of $Ca^{2+}$ dynamics[16], and several strategies for a fine determination of $[Ca^{2+}]_{in}$ by a classical fluorometry[17], fluorescence lifetime imaging (FLIM)[18,19], and recently reported photochromism-based quantification[20] have been developed. These methods commonly utilized GECIs with relatively low $Ca^{2+}$ affinity ($K_d > 90$ nM), and differently predicted the resting $[Ca^{2+}]_{in}$ from 50 nM[21] to

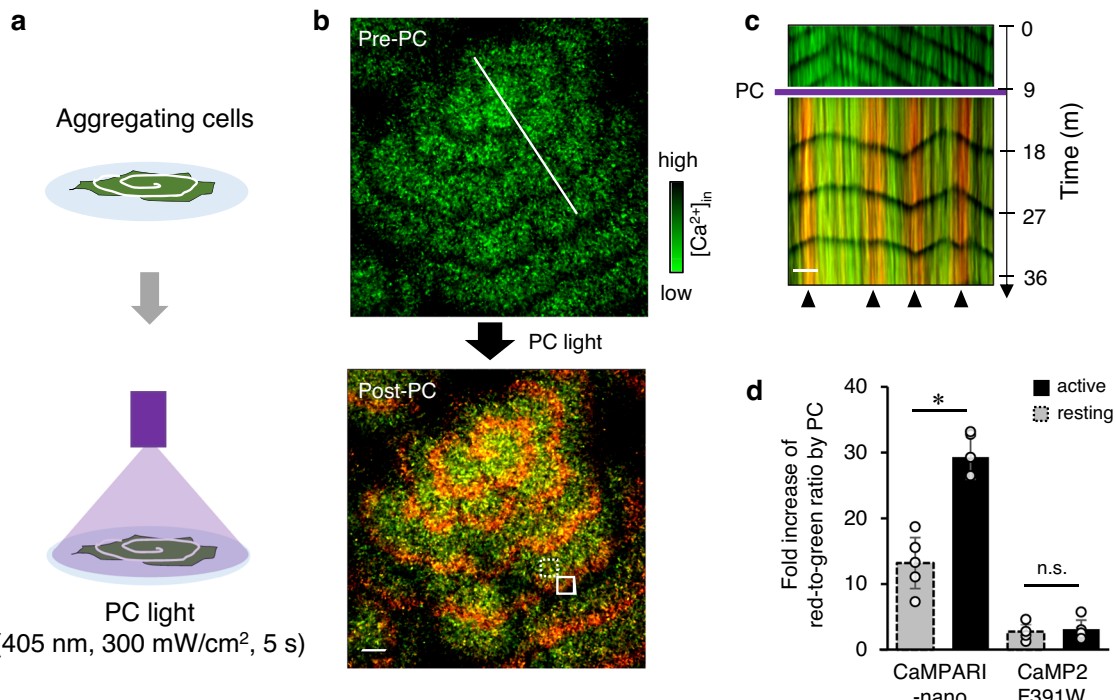

**Fig. 3 | Functional highlighting of actively signaling cells by CaMPARI-nano. a** A scheme for the irreversible photoconversion of cells with $[Ca^{2+}]_{in}$ transients at low nM concentrations. Cells expressing CaMPARI-nano lacking the reference marker were cultured on a non-nutrient agar plate to spontaneously generate spiral-shaped signaling waves of $[Ca^{2+}]_{in}$. **b** Composite images of the green and red form of CaMPARI-nano at pre- and post-photoconversion. **c** Kymograph at the line in b shows irreversible highlighting of actively signaling cells at the timing of photoconversion (arrowheads). **d** Photoconversion contrast in active (solid box in **b**) and resting (dashed box in **b**) areas containing ~10 cells. Average of five ROIs with s.d.. The asterisk indicates $p < 0.05$. n.s., not significant. Scale bar: 0.1 mm.

110 nM[22] for HEK293 cells and 15 nM[23] to 50 nM[21] for HeLa cells. Attributing inconsistent estimation of resting $[Ca^{2+}]_{in}$ to insufficient detection sensitivity at a low nM range, we evaluated the relative levels across different cell types. This was achieved by comparing the fluorescence intensities of CaMPARI2 variants, which span low-to-ultrahigh $Ca^{2+}$ affinities. CaMPARI2 variants, when co-expressed with the reference marker P2A-mCherry, exhibited uniform cellular distribution as shown in Supplementary Fig. 9a. This observation indicates that RS20-mutations and topology mutation do not adversely affect the expression of these indicators. In HEK293 cells, treatment with BAPTA-AM or ionomycin resulted in CaMPARI2 fluorescence ratio (*R*: GECIs/reference) of 6.17 for $Ca^{2+}$ depletion and 0.20 for $Ca^{2+}$ saturation. This provided a substantial dynamic range of 5.97, as detailed in Supplementary Fig. 9b. In non-treated HEK293 cells, CaMP2_F391W (in vitro $K_d$ = 121 nM) and CaMPARI-nano (in vitro $K_d$ = 19 nM) showed fluorescence ratio of 5.4 and 0.25, respectively, as shown in Supplementary Fig. 9b. This resulted in fractional changes for each indicator of 13 and 99%, respectively, detailed in Supplementary Fig. 9c. The fractional change for CaMP2_F391W+3mut was 64% (*R* = 2.3; Supplementary Fig. 9c), indicating that the resting $[Ca^{2+}]_{in}$ in HEK293 cell is approximately 50 nM, slightly above the in vitro $K_d$ value of 40 nM for CaMP2_F391W+3mut. Similar results were obtained for HeLa cells with smaller fractional changes for three variants of CaMPARI2 (Supplementary Fig. 9c), indicating that HeLa cells have a slightly lower resting $[Ca^{2+}]_{in}$ than HEK293 cells. Glioma-derived U-87 cells exhibited even lower resting $[Ca^{2+}]_{in}$ levels, with CaMP2_F391W+3mut and CaMPARI-nano showing fractional changes of 23% and 94%, respectively (*R* values of 4.8 and 0.32, as detailed in Supplementary Fig. 9b, c). Thus, the resting $[Ca^{2+}]_{in}$ in U-87 cells was estimated to be approximately 30 nM, which lies between the $K_d$ values of 43 nM and 19 nM. Primary cultures of astro-glia showed a very similar pattern of fractional change as observed in U-87 cells (Supplementary Fig. 9c), suggesting that a low resting $[Ca^{2+}]_{in}$ is likely a common

characteristic among glia-related cells. While resting $[Ca^{2+}]_{in}$ levels can be determined more precisely using sophisticated quantification methods[18,20,23], a set of GECI, including high-affinity variants, offers practical approach for rapid estimation and comparison of low nM range $[Ca^{2+}]_{in}$ levels across various cell types.

## Irreversible labeling of actively signaling cells by CaMPARI-nano

CaMPARIs are effective tools for optically highlighting firing neurons and their circuits since they can irreversibly label cells displaying $[Ca^{2+}]_{in}$ transients via green-to-red photoconversion (PC) upon UV light irradiation[10]. In in vitro experiments, CaMPARI-nano showed comparably high photoconversion efficacy to the parental CaMP2_F391W under $Ca^{2+}$-saturated conditions (Fig. 1c; Supplementary Fig. 4a). Given its increased affinity for $Ca^{2+}$ (Figs. 1 and 2), CaMPARI-nano enables the functional labeling of cells showing physiological $[Ca^{2+}]_{in}$ transients in the low nanomolar range.

Thus, we attempted to label the actively signaling cells in the *D. discoideum* population that showed $[Ca^{2+}]_{in}$ transients below 100 nM by utilizing the photoconversion of CaMPARI-nano. We applied PC light (405 nm, 300 mW cm$^{-2}$, 5 s) to the entire population of nutrient-starved *D. discoideum* cells expressing CaMPARI-nano during the development of spiral-shaped signaling waves of $[Ca^{2+}]_{in}$[6] (Fig. 3a). A red spiral pattern was observed, which was identical to the spatial distribution of cells showing $[Ca^{2+}]_{in}$ transients at the time of PC (Fig. 3b, c; Supplementary Movie 2). The red-to-green intensity ratios of active and resting cells after PC were 29.3 and 13.2, respectively (Fig. 3d left). This indicates that the PC of CaMPARI-nano efficiently highlights cells showing $[Ca^{2+}]_{in}$ transients with high contrast (2.2-fold difference). On the other hand, cells expressing parental CaMP2_F391W showed red-to-green intensity ratios of 3.1 and 2.8 in the active and resting states, respectively (Fig. 3d right; Supplementary Fig. 10), resulting in a low contrast on the PC (1.1-fold difference). These results demonstrate the advantage of CaMPARI-nano in highlighting cells

**Fig. 4 | Triple-function imaging of Ca²⁺, cGMP, and cAMP. a** Schematic illustration of triple-function imaging in cells simultaneously expressing indicators for Ca²⁺, cGMP, and cAMP. **b** Fluorescence signal changes of BGECO-nano, Green cGull, and R-FlincA during three cycles of wave propagation. The 3-frame-moving averaged data in ROI containing ~10 cells on a non-nutrient agar plate was shown. Open and closed arrowheads indicate the timing of rise and peak. **c** A close-up view of the signaling wave at the traveling front (box in **b**). Scale bar: 0.1 mm.

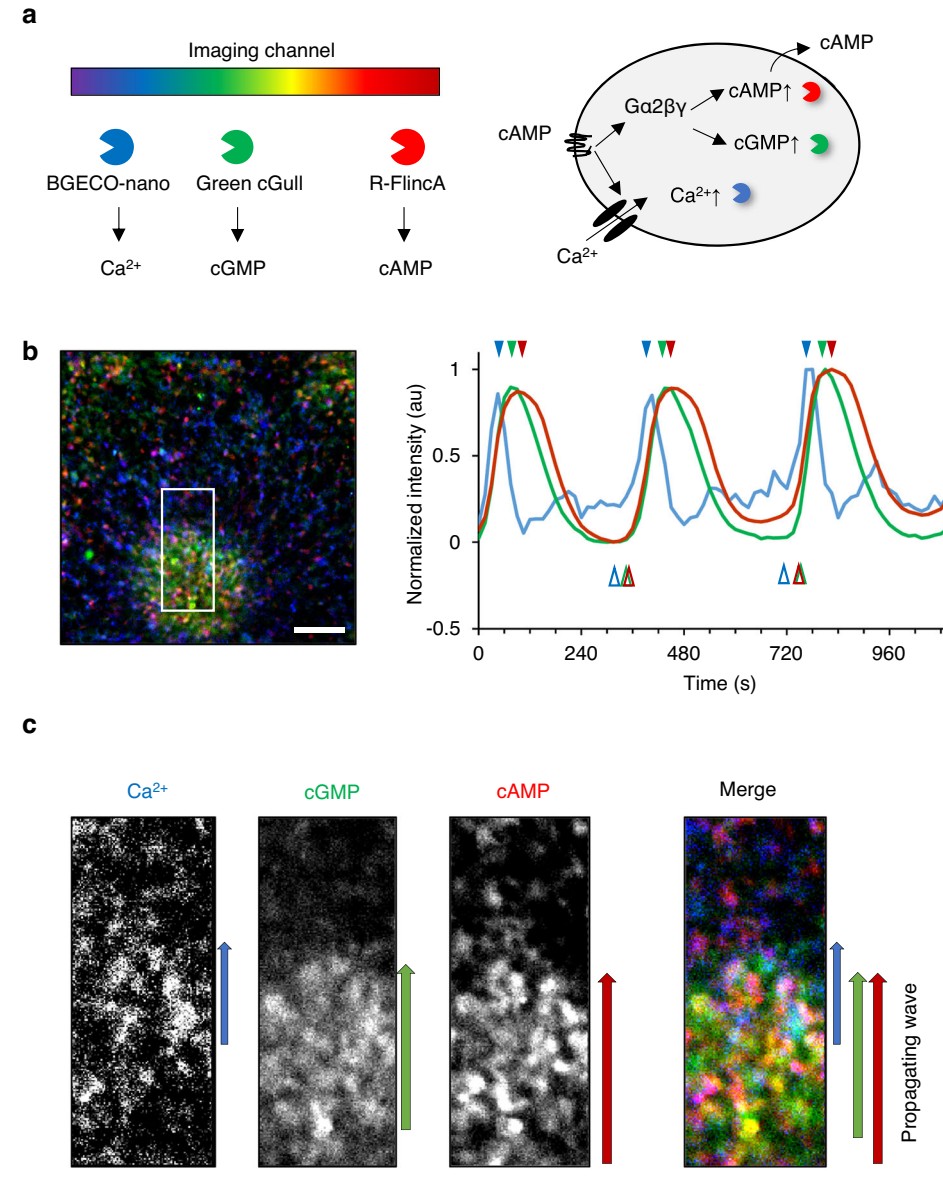

---

with [Ca²⁺]ₗₙ transients below 100 nM, where previous CaMPARIs were not applicable.

## Triple-function imaging of intracellular Ca²⁺, cGMP, and cAMP

Next, we validated the utility of color variants of ultrahigh-affinity GECIs by simultaneous imaging of multiple signaling events. We focused on the intracellular dynamics of Ca²⁺, cAMP, and cGMP (cyclic guanosine monophosphate), which are common second messengers regulating gene expression, cellular movements, and metabolism in various cells[24]. During the chemotactic aggregation of *D. discoideum* cells, these second messengers ([Ca²⁺]ₗₙ, [cAMP]ₗₙ, and [cGMP]ₗₙ) transiently increase in response to spontaneously generated [cAMP]ₑₓ[25–27]. Although it has been speculated that [cGMP]ₗₙ rises first, followed by [Ca²⁺]ₗₙ and [cAMP]ₗₙ based on independent measurements using different methods[25,28], their activation sequences have not been validated due to the technical challenges of simultaneous multi-functional imaging.

To investigate the activation sequence, we obtained *D. discoideum* cells expressing BGECO-nano (blue, Ca²⁺), Green cGull (green, cGMP)[29], and R-FlincA (red, cAMP)[30], and analyzed the dynamics of these signals during chemotactic aggregation (Fig. 4a). Triple-channel imaging revealed that [Ca²⁺]ₗₙ showed the earliest rise and peak timings during repeated signaling events (Fig. 4b blue; Supplementary Movie 3). [cGMP]ₗₙ and [cAMP]ₗₙ

showed delayed initiation and peak timings (Fig. 4b green and red, respectively) compared to [Ca²⁺]ₗₙ. The spatial pattern of the three signals also reflected these temporal patterns: [Ca²⁺]ₗₙ signals were located at the front of the propagating waves, while [cGMP]ₗₙ and [cAMP]ₗₙ signals were found in the middle-to-tail of the propagating waves, with large overlap (Fig. 4c; Supplementary Movie 3).

These direct observations confirmed that the activation sequence of the three signaling pathways started with [Ca²⁺]ₗₙ followed by [cGMP]ₗₙ and [cAMP]ₗₙ, demonstrating the utility of the expanded color hue of high-affinity GECIs in analyzing the temporal relationship of multiple signals.

## Affinity tuning of other 1FP-type indicators

To conceptually examine whether linker length optimization in topology mutant is useful for affinity tuning of other 1FP-type indicators, we focused on the glutamate indicator iGluSnFR[31] and the potassium ion (K⁺) indicator GINKO1[32]. iGluSnFR, with split sensor motifs at the head and tail of cp-GFP, was subjected to circular permutation to link its sensor motifs (Fig. 5a). As reported previously[33], the recombinant protein of topology mutant incorporating a long linker (24 a.a.) showed a slightly increased affinity to glutamate ($K_d = 36$ nM) than the parental construct ($K_d = 40$ nM) (Fig. 5b). Linkers of 12, 6, and 3 a.a. further increased affinity reaching to $K_d$ of 6.2 nM with 3 a.a. linker (Fig. 5c; Supplementary Table 2), indicating that the linker

**Fig. 5 | Affinity tuning of other 1FP-type indicators by linker length optimization in topology mutant. a** Molecular design of parental iGluSnFR and its variants. In vitro titration curve (**b**) and $K_d$ of purified protein (**c**). Estimated $K_d$ value was plotted with uncertainty (error bars) in Hill fitting of the averaged trace from three independent measurements.

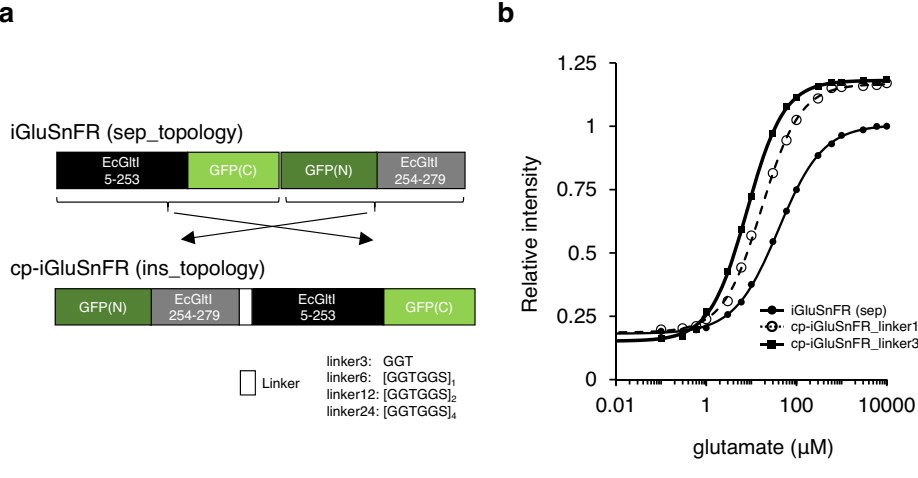

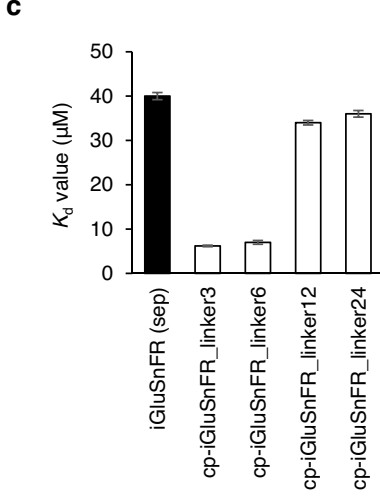

length optimization in topology mutant is applicable to 1-FP type indicator other than GECIs. The biochemical properties of cp-iGluSnFR_linker3 showed 1.4-fold increased signal change ($\Delta F/F_0$) associated with both dimmer baseline and brighter peak intensity, while p$K$a and Hill coefficient were unaffected. In HEK293cell, the surface-displayed indicators by using a modified min-Display vector showed no cellular aggregates and unaffected brightness with large signal changes upon glutamate stimulation (Supplementary Fig. 11), suggesting their potential applicability in living cells.

In the K$^+$ indicator GINKO1, originally having an insertion topology, the linker length variations did not increase the affinity (Supplementary Fig. 12), indicating that the strategy is not applicable to all 1FP-type indicators. Still, the topology mutant having the sensor separated topology (cp-GINKO1) showed decreased affinity, demonstrating the wider applicability of circular permutation in affinity tuning of 1FP-type indicators.

## Discussion

In this study, we developed a series of 1FP-type GECIs with greatly increased Ca$^{2+}$ affinity by linker length optimization in topology mutants in combination with classical mutagenesis. Utilities of developed GECIs were demonstrated for analyzing physiological [Ca$^{2+}$]$_{in}$ in living samples having a low resting [Ca$^{2+}$]$_{in}$ and a small amplitude of [Ca$^{2+}$]$_{in}$ transients at a low nM range.

To what extent can this engineering strategy be broadly utilized in the development of indicators? One might question why certain GECIs with the insertion topology did not exhibit ultrahigh-affinity. Examples include camgaroo[11] and NEMO[34]. In contrast to here examined GECIs, camgaroo features a single Ca$^{2+}$-sensing motif (CaM) with approximately one order of magnitude lower Ca$^{2+}$ affinity compared to CaM-M13. Therefore, it is

reasonable for camgaroo not to display ultrahigh-affinity. Recently reported NCaMP7[35] and its single mutation derivative NEMO[34], both incorporating an insertion of CaM-RS20 in mNeonGreen, exhibit quite similar molecular organization to here tested GECIs. NCaMP7[35] and NEMOs share 14 mutations on CaM, including affinity-lowering ones (*e.g.*, N53I[36]). Hence, there is a possibility that the inherently low Ca$^{2+}$ affinity of NCaMP7/ NEMO could be increased by substituting mutated CaM with intact ones. Unfortunately, the linker length optimization did not work for GINKO1 as reported for the FRET-based K$^+$ indicator, GEPII[37]. We have no clear reasoning for this, but the optimal linker length for the highest affinity varied in the four tested indicators (12 a.a./CaMPARI-nano, 6 a.a./BGECO-nano and RCaMP-nano, 3 a.a./cp-iGluSnFR), suggesting that the spatial relationship between two sensor motifs would have significance and indicators with closely positioned sensor motifs would be more compatible with our strategy. Besides these potential limitations, we believe that affinity tuning by the linker length optimization would be applicable to a variety of 1FP-type indicators other than GECIs as supported by iGluSnFR.

The mechanisms behind achieving ultrahigh-affinity are intriguing. In principle, high-affinity can be attained through slowed off- and/or accelerated on-kinetics. While many GECIs achieve high-affinity solely by slowing down off-kinetics as demonstrated in CaMPARI[10], the ideal GECIs should involve fast off- and even faster on-kinetics, especially to accommodate rapid dynamics like neural activities. In GECIs, such ideal mutations leading to high-affinity are rarely identified, while mutations leading to slow-off can be found in the sensor motifs[10]. Therefore, optimizing the linker length in topology mutants is crucial for primarily accelerating on-kinetics with slightly affecting off-kinetics in CaMPARI2s or, rather, improving off-kinetics in RCaMPs. The reason why R-GECO[13] derived RCaMPs[14] solely

could achieve the ideal high-affinity, namely a fast off- and an even faster on-kinetics, is not certain, but similar effects can be expected in other indicators. Moreover, it might be possible to re-design existing GECIs to accommodate improved kinetics but a normal $Ca^{2+}$ affinity. For example, while cp-CaMP2_linker12 achieved a $K_d$ of 50 nM by slowed off- and accelerated on-kinetics, further introducing an affinity-lowering mutation to speed up off-kinetics[10] could lead to the development of an indicator with normal affinity ($K_d \sim 100$ nM) possessing both faster on- and off-kinetics compared to the parental CaMP2_F391W. This could potentially enable a more precise photoconversion-based highlighting of actively signaling cells with higher temporal resolution.

Collectively, the linker length optimization in topology mutants would benefit the development of 1FP-type indicators for an improved affinity and faster kinetics, which, in turn, deepen our understanding of signaling dynamics of cell populations in a wide range of life sciences, including neuroscience, developmental biology, and physiology.

## Methods

### Molecular biology
cDNAs encoding CaMPARI1, B-GECO1.0, R-GECO1.2, iGluSnFR, and GINKO1 were obtained from Addgene. CaMPARI2 and R-CaMP1.01 were reconstructed from CaMPARI1 and R-GECO1.2, respectively, using PCR-based modifications. Circularly permuted constructs were generated by assembling PCR amplicons using the In-Fusion HD Cloning Kit (TaKaRa). Sequence-verified cDNAs were cloned into a pRSET$_B$ vector. For the expression of indicators in *D. discoideum* cells, codon usage was optimized for *D. discoideum* cells by a gene synthesis service (Eurofins Genomics) because non-optimized cDNAs were not expressed. These cDNAs were cloned into expression vectors for *D. discoideum* cells (pDM304, 358) as previously described[30].

### Screening of high-affinity mutations in RS20
CaMP2_F391W cDNA in pRSET$_B$ vector was subjected to site-directed PCR mutagenesis to introduce the representative amino acid with a side chain of basic, acidic, non-polar/uncharged, hydrophobic, and others. Proteins were expressed in *Escherichia coli* (JM109(DE3)) at 37 °C overnight. Cells were pelleted and chemically lysed with B-PER (Thermo Fisher Scientific). Fluorescent intensities were measured for aliquots of cleared supernatants at four different $Ca^{2+}$ concentrations to roughly evaluate the increase or decrease in $Ca^{2+}$ affinity.

### Protein purification and in vitro spectroscopy
Recombinant indicators, N-terminally tagged with polyhistidine, were expressed in *E. coli* (JM109(DE3)) for 4–5 days at 22 °C with gentle shaking at 130 rpm. The *E. coli* cells were lysed using B-PER. Subsequently, the recombinant indicator was purified using a Ni-NTA column (Wako) and underwent buffer exchange employing a Microsep Advance Centrifugal Device with a 30 K MWCO (PALL). The buffers used for the purified indicators were exchanged with the appropriate ones for GECIs[6], iGluSnFR[31], and GINKO[32]. Protein concentrations were determined as previously reported[13]. Fluorescence spectra were recorded using a fluorescence spectrophotometer (F-7000, Hitachi), and absorption spectra were recorded using a microplate reader (SpectraMax iD3, Molecular Devices). For kinetic analysis, the dissociation dynamics were measured using a stopped-flow device (SFS-853 and FP-8000, JASCO), as previously reported[9]. Briefly, fluorescence signal changes by depletion of $Ca^{2+}$ from the indicators through rapid mixing with 10 mM EGTA were recorded every 5 ms up to 30 s, then were fitted to a single exponential curve to determine the $k_{off}$ value using OriginPro 9 software (LightStone). The $k_{on}$ value was calculated using the independently determined $K_d$ through the following equation: $k_{on} = k_{off} K_d^{-1}$.

### Titration and photoconversion
$Ca^{2+}$, glutamate, $K^+$, and pH titrations were performed as described[6,31,32]. Briefly, the normalized fluorescence intensities of the GECIs (BGECOs,

446 nm; CaMPARIs, 515 and 574 nm; RCaMPs, 589 nm) and other indicators (iGluSnFRs and GINKOs, 515 nm) at different analyte concentrations were fitted to Hill plots using OriginPro 9 software. The photoconversion of CaMPARIs and their analysis was performed as described[9]. Briefly, 150 nM of recombinant indicators at different $Ca^{2+}$ concentrations were prepared in a 96-well glass-bottom plate and photoconverted by illuminating with 405 nm LED light (Prizmatix) at 250 mW cm$^{-2}$. For the time-course analysis, replicates of samples with different illumination times (0–150 s) were prepared. Red fluorescence was measured after free $Ca^{2+}$ was removed by adding 5 mM EGTA. The changes in red fluorescence were fitted to a single exponential curve to obtain photoconversion rate constants at different $Ca^{2+}$ concentrations. For the $Ca^{2+}$ titration of the red form of CaMPARIs, recombinant indicators (30 μM, 100 μl) were prepared in a 96-well glass-bottom plate at saturating $Ca^{2+}$ levels. They were then illuminated with 405 nm LED light (250 mW cm$^{-2}$). Samples were gently mixed every 10 s, and a 1-μl of aliquot was checked for the red-to-green fluorescence intensity ratio. After saturation with the emergence of the red form, the samples were collected and exchanged in a $Ca^{2+}$-free buffer for $Ca^{2+}$ titration.

### Culture, transformation, and imaging of mammalian cells
HeLa (RCB0007) and HEK293 (RCB1637) cells were obtained from RIKEN BRC and were cultured in DMEM supplemented with 10% FBS. U-87 (HTB14) cell obtained from ATCC was cultured in RPMI with 10% FBS. The pCMV expression vector encoding cDNAs of GECIs followed by P2A_mCherry were transfected into cells on a 35-mm glass bottom dish coated with Poly L-Lysin by using FuGENE-HD (Promega) according to the manufacturer's instruction. Live cell imaging was performed using an inverted confocal microscope (Nikon A1R, Nikon) equipped with a PlanApo20× objective (0.75 N.A., Nikon), a 488-nm DPSS laser (20 mW, Melles Griot) for green-form CaMPARIs, and a 561-nm DPSS laser (25 mW, Melles Griot) for mCherry band-pass together with emission filters of 500–550 nm (green), and 570–620 nm (red). For a saturation and depletion of cellular $[Ca^{2+}]$, cells treated with 1 μM Ionomycin (MERCK) and 15 μM BATPA-AM (Selleckchem) for 10 min were then imaged by a confocal microscope. For a cell surface expression of iGluSnFRs, their cDNAs were cloned into a modified pMinDisplay vector[31,33], in which a spacer was newly inserted between Igk and iGluSnFR for efficient processing with signal peptidase. In-cell titration was performed as reported previously[31,33]. All assay was performed at 48 h after the transfection. Primary cultures of astro-glia were prepared by differentiation of neurosphere prepared from the ganglionic eminence at embryonic day 18.5 C57BL/6 J mice, as described previously[38]. Briefly, neurospheres were dissociated into single cells and plated onto 35-mm glass bottom dishes. These cells were cultured in Neurobasal medium supplemented with LIF and BMP-2 for astrocytes differentiation. One day after differentiation, cells were transfected with GECIs expression vectors using Lipofectamin 2000 (Thermo Fisher Scientific). Imaging was conducted at four days after DNA transfection. All experimental procedures using mice were performed in accordance with the ethical guidelines of Tokushima University, and this study was approved by the Animal Research Committee, Tokushima University.

### Cell culture, transformation, and imaging of *D. discoideum* cells
The axenic strain of *D. discoideum*, Ax2, was cultured and transformed following the methods described[39,40]. For transformation, the cells were washed and suspended in ice-cold EP buffer (6.6 mM $KH_2PO_4$, 2.8 mM $Na_2HPO_4$, and 50 mM sucrose, pH 6.4) at $1 \times 10^7$ cells ml$^{-1}$. An 800-μl cell suspension mixed with 10 μg of expression plasmids in a 4-mm-wide cuvette was subjected to electroporation (two 5-s separated pulses with 1.0 kV and a 1.0 ms time constant) using a MicroPulser (Bio-Rad). These cells were then plated in 90-mm plastic dishes with HL5 medium and incubated at 22 °C for 18 h under non-selective conditions. Subsequently, the cells were cultured in the presence of 10 μg ml$^{-1}$ G418 (Wako) for pDM304 or 35 μg ml$^{-1}$ hygromycin (Wako) for pDM358. After 4–7 days, colonies showing higher expression of the indicator and lower heterogeneity

were manually selected. To initiate chemotactic aggregation of *D. discoideum*, cells cultured in HL5 medium were washed with development buffer (DB: 5 mM $Na_2HPO_4$, 5 mM $KH_2PO_4$, 1 mM $CaCl_2$, and 2 mM $MgCl_2$ at pH 6.4). For on-agar development, $2.5 \times 10^6$ cells $cm^{-2}$ were plated on 1% agar (5 mM MES, 5 mM $CaCl_2$, 5 mM $MgCl_2$, pH6.5) supplemented with 1.5 mM caffeine, and cultured for 6 h at 22 °C. Live cell imaging was performed using an inverted confocal microscope (Nikon A1R, Nikon) equipped with a PlanApo10× objective (0.45 N.A., Nikon), a 405-nm diode laser (36 mW, Melles Griot) for BGECOs, a 488-nm DPSS laser (20 mW, Melles Griot) for green-form CaMPARIs, and a 561-nm DPSS laser (25 mW, Melles Griot) for R-FlincA and red-form CaMPARIs. The following band-pass emission filters were used: 425–475 nm (blue), 500–550 nm (green), and 570–620 nm (red). The hardware was controlled using Nikon NIS-Elements software (Nikon), and image processing was performed using Fiji software (http://fiji.sc/Fiji). Photoconversion of CaMPARIs was performed on cells on 1% agar by illuminating with 405 nm light (300 mW $cm^{-2}$, 5 s). For a photobleaching experiment, cells expressing GECIs were continuously illuminated with LED at >100-fold higher power of excitation light (Nikon) than that in normal $Ca^{2+}$ imaging. Wide-field images were captured by sCMOS camera (ORCA-Fusion BT, Hamamatsu photonics) every 0.1 s for 5 min.

## Image analysis and quantification
The calculation of the magnitude of signal change in Fig. 2c is detailed visually in Supplementary Fig. 8 and described in its legend. To analyze the dynamics of $Ca^{2+}$, cAMP, and cGMP in *D. discoideum*, the acquired images were imported into Fiji software for further analysis. Following background subtraction, the intensities of the appropriate channels were analyzed for each ROI, either on a single timeframe or as a time series. The collected data were analyzed using Microsoft Excel.

## Statistics and reproducibility
Nonparametric statistics were used to analyze the data. All statistical tests were two-tailed. The analysis was performed using *R* statistical environment version 4.2.0 (https://www.r-project.org/). Experiments were repeated at least 3 times except for Fig. S4a (N = 1), Fig. S8b (N = 2), Fig. S11 (N = 2).

## Data availability
The data used in this study are available upon reasonable request. GenBank accession number for CaMPARI-nano, BGECO-nano, and RCaMP-nano is OR079885, OR079886, OR079887, respectively. The source data behind the graphs in the paper can be found in the Supplementary Data file.

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

## Acknowledgements

This work was supported by a Grant-in-Aid for Scientific Research on Innovative Areas "Singularity Biology (No. 8007)" (18H05408 to T.N., 18H05415 to K.H.), Research Program of "Five-star Alliance" in "NJRC Mater. & Dev." (T.N. and K.H.), and Japan Science and Technology Agency Moonshot R&D grant (JPMJMS2025 to A.S.)

## Author contributions

K.H. and T.N. designed and supervised the project with contributions from Y.H. A.I. T.M. A.S. T.S. All experiments and data analyses were carried out by Y.H. A.I. A.S., and T.M.. Y.H. T.N., and K.H. wrote the manuscript; all authors reviewed the manuscript.

## Competing interests

The authors declare no competing interests.
