## [Peer Review File · Communications Biology]

Reviewers' comments:

Reviewer #1 (Remarks to the Author):

Hara et al. reported the generation of inverted versions of several GECIs with improved Ca affinities. The developed sensors were tested in mold cells, known for Ca transients under 100 nM amplitudes. The authors demonstrated that the topology inversion of other sensors, such as potassium and glutamate, can also be used to modulate affinities for the target ligands. However, I have concerns about the novelty and significance of this study. First of all, inversion of the GECI topology to modulate affinity is not a novel approach: it was demonstrated more than 5 years ago that insertion of CaM/M13 pair into FPs can improve Ca affinity (see Qian et al, A Bioluminescent Ca²⁺ Indicator Based on a Topological Variant of GCaMP6s, 2018); moreover, several recently published GECIs exhibit Ca affinities comparable to that reported in this study while having larger fluorescence dynamic range, for example, jGCaMP8s (K_d=46 nM) or recent GECI iNTnC2 from Subch (K_d=49 nM), which should have similar performance in live cells. Therefore, stating in the Abstract that it is “a breakthrough strategy” is inadequate. Fine-tuning the affinity of GINKO and iGluSnFR seems to be more innovative. However, the authors did not demonstrate the applicability of the generated variants. Overall, the presented results did not convince me that the present sensors may find broad immediate applications.

1. To demonstrate the beneficial properties of CAMPARI-nano the authors should compare it side-by-side with jGCaMP8s and iNTnC2 (assessment of other biochemical properties is crucial here, including brightness, photostability, kinetics). Furthermore, the authors should acknowledge previous studies reporting the development and characterization of topologically inverted GECIs as well as GECIs with high affinity, as mentioned above (it might be even worse to include them in Supplementary Table 1).
2. The introduction of the manuscript is misleading. The authors listed several biological systems with low Ca, including “malaria parasites³, plant cells⁴, and murine astrocytes⁵” however in Results they used mold cells (which do not fall into any of the listed categories above) for the demo. I would strongly suggest using one more model to demonstrate the applicability of the developed sensors side-by-side with jGCaMP8s or iNTnC2. Since the major point of the manuscript is the new GECIs with nanomolar Ca affinities, I would recommend taking a more systematic approach and listing all known cell types and organisms with [Ca] in the lower nanomolar range and presenting it in the form of a table in the main text. It will help the reader appreciate the importance of this work more.
3. It is unclear why the range of 20-30 nM was selected as the target. Please provide clear biological justification in the text. Again, Table with all biological systems with a known lower nM [Ca] range may help make the point here.
4. Further characterization of cpGINKO and cpiGluSnFR is lacking here, it is hard to judge the significance of these sensors with the assessment of other properties, which are equally important to K_d for applicability.

Reviewer #2 (Remarks to the Author):

The paper focuses on developing ultrahigh-affinity single-fluorescent-protein-based GECIs for live imaging of low nanomolar range calcium dynamics. The researchers employed circular permutation to

increase the affinity of existing GECIs while preserving their basic properties. The resulting GECIs, CaMPARI-nano, BGECO-nano, and RCaMP-nano, have a K_d of 20-30 nM, enabling unique biological applications such as detecting low nanomolar Ca^{2+} dynamics and multi-functional imaging with other second messengers. The study also demonstrated the effectiveness of the circular permutation strategy in modulating indicator affinity, which could be applied to other single-fluorescent-protein-based indicators of glutamate and potassium. I have the following concerns:

1. It should be noted that circular permutation of GECIs with fluorescent proteins in the circularly permuted topology is equivalent to the insertion of sensory domains into fluorescent proteins in the WT topology, which has been previously reported. While there is novelty in this paper, the topology itself is not new. The authors should acknowledge such a fact.
2. Related to the previous point, previous Ca^{2+} sensors have been developed by inserting calcium sensory domains into fluorescent proteins in the WT topology. In fact, one of the very first calcium sensors was made by inserting CaM into GFP in 1990s (PNAS 1999). The authors should review these sensors in the Introduction. It is a bit confusing that these previously reported Ca^{2+} sensors didn't show ultralow calcium affinity (e.g., <https://www.nature.com/articles/s41592-023-01852-9>). The authors should discuss the reasons for discrepancies in their affinity compared to the ultralow affinity achieved in this study.
3. The authors used 6 aa and 12 aa linkers for calcium sensors, but only used the 6-aa linker for other sensors. The reasoning behind this decision was not explained.
4. Further investigation is needed to understand how topology change affects the properties of the sensors. The authors could compare the initial constructs and new constructs with different linker lengths for k_{on} , k_{off} , pK_a , and thermostability.
5. Error bars should be added to Figure 2 and Figure S5, and a statistical analysis should be performed to show the superiority of CaMPARI-nano over CaMP2 (along with dead mutants and YC-Nano).
6. It would be interesting to investigate the application of this circular permutation strategy to later GCaMP variants (GCaMP6 and above).
7. The fact that the cpCamP_link6 sensor has almost the same K_d as the original construct suggests that linker optimization may be more important than circular permutation for increasing affinity.
8. "However, this strategy has not been considered for 1FP-type GECIs because most..." and "we employed a previously unexplored strategy...". Again, many previous sensors were based on the same topology (insertion of the sensory domain into fluorescent proteins). Other papers didn't show that the insertion topology led to higher affinity. Would the phenomenon be largely caused by linker optimization here rather than the circular permutation itself?
9. The authors should provide more information on the sensors in Figure 5, including the dynamic change and whether the linkers were optimized for these sensors.

Reviewer #3 (Remarks to the Author):

In this manuscript, Yusuke et al. reported a strategy that combines biosensor topology change with rationally designed sensing domain mutations to engineer high-affinity GECIs. The authors also extended the topology-altering approach to glutamate and potassium biosensors to demonstrate the applicability of this method.

Overall, I find this manuscript suitable for publication in *Communications Biology*. However, to better inform the readers, I believe the following concerns should be addressed.

The authors extensively used the combination of three affinity-enhancing RS20 mutations in this work. These mutations effectively decrease the GECI K_ds alongside the circular permutation strategy, yet the details of how these mutations were identified need to be included. I recommend the following changes:

1. Provide detailed information on the screening of the nine mutations, preferably with a rationale for why these particular mutations were chosen.
2. Specify the three key mutations in the main text when they are mentioned (e.g., Line 88, 107).
3. Include the site-directed mutagenesis protocol and screening protocol in the Methods section.
4. Considering the importance of these mutations, I suggest changing the title to "Affinity tuning of single fluorescent protein-type indicators by circular permutation and rational mutagenesis"

In the current manuscript, it is unclear to me whether the affinity change in BGECO-nano and RCaMP-nano is solely attributed to the circular permutation, the three affinity-enhancing RS20 mutations, or the combination of both strategies. Therefore, it is of utmost importance to show the calcium K_d titration curves and report the K_d values for cpB-GECO1 and cpR-CaMP1.01 (Line 129).

On Line 24 and Line 251, the authors described their strategy as a "breakthrough" and "unexplored." However, to my knowledge, GECI topology change has been reported before in the paper by Qian, Y., Rancic, V., Wu, J., Ballanyi, K., & Campbell, R. E. (2018). "A Bioluminescent Ca²⁺ Indicator Based on a Topological Variant of GCaMP6s." *ChemBioChem*, 20(4), 516–520. I recommend that the authors properly cite this paper and edit the wording in the current manuscript accordingly.

We appreciate valuable comments by reviewers. According to their suggestions, we largely revised the manuscript including the title by focusing on the effects of “newly performed linker length optimization” on “previously reported topology mutants.” Also, we carefully responded to all the concerns and discussed the novelty of our findings on the basis of comparative and quantitative studies as follows:

Reviewer #1 (Remarks to the Author):

Hara et al. reported the generation of inverted versions of several GECIs with improved Ca affinities. The developed sensors were tested in mold cells, known for Ca transients under 100 nM amplitudes. The authors demonstrated that the topology inversion of other sensors, such as potassium and glutamate, can also be used to modulate affinities for the target ligands. However, I have concerns about the novelty and significance of this study. First of all, inversion of the GECI topology to modulate affinity is not a novel approach: it was demonstrated more than 5 years ago that insertion of CaM/M13 pair into FPs can improve Ca affinity (see Qian et al, A Bioluminescent Ca²⁺ Indicator Based on a Topological Variant of GCaMP6s, 2018); moreover, several recently published GECIs exhibit Ca affinities comparable to that reported in this study while having larger fluorescence dynamic range, for example, jGCaMP8s (K_d=46 nM) or recent GECI iNTnC2 from Subch (K_d=49 nM), which should have similar performance in live cells. Therefore, stating in the Abstract that it is “a breakthrough strategy” is inadequate. Fine-tuning the affinity of GINKO and iGluSnFR seems to be more innovative. However, the authors did not demonstrate the applicability of the generated variants. Overall, the presented results did not convince me that the present sensors may find broad immediate applications.

Q1-1. To demonstrate the beneficial properties of CaMPARI-nano the authors should compare it side-by-side with jGCaMP8s and iNTnC2 (assessment of other biochemical properties is crucial here, including brightness, photostability, kinetics). Furthermore, the authors should acknowledge previous studies reporting the development and characterization of topologically inverted GECIs as well as GECIs with high affinity, as mentioned above (it might be even worse to include them in Supplementary Table 1).

A1-1. We agree with this comment.

- (1) In the introduction, we included the previously developed GECIs having high affinity (jGCaMP8s) and increased Ca²⁺ affinity in the topology mutant of GCaMP6 (Qian et al, 2018). [L70-71 for jGCaMP8s, L78-L85 for topology mutant of GCaMP6]**
- (2) We performed a detailed analysis of *in vitro* properties for CaMPARI-nano, including jGCaMP8s (Supplementary Table 1). The data showed that CaMPARI-nano has a higher Ca²⁺ affinity (×2) and significantly slower dissociation kinetics (×1/18) compared to jGCaMP8s (Fig. 1f). The brightness (Supplementary Fig. 8a), photostability (Supplementary Fig. 8b), signal change (Fig. 2c), SNR (Fig. 2d), and kinetics (Fig. 2e) were also analyzed in living cells, clarifying both advantages (3-fold larger signal change, Fig. 2c and Supplementary Fig. 8a; 2-**

fold large SNR, **Fig. 2d**) and disadvantages (8-fold slower decay kinetics, **Fig. 2e**) of CaMPARI-nano over jGCaMP8s in the detection of Ca²⁺ signaling in *Dictyostelium discoideum*. These analyses were expanded to BGECO and RCaMPs to show that RCaMP-nano has faster or comparable dissociation kinetics to that of jGCaMP8s in *in vitro* (**Fig.1f**) and *in cell* (**Fig. 2e**), respectively.

To reflect the above findings, the result part [L154-170, and 197-207] has been largely revised.

Q1-2. The introduction of the manuscript is misleading. The authors listed several biological systems with low Ca, including “malaria parasites³, plant cells⁴, and murine astrocytes⁵” however in Results they used mold cells (which do not fall into any of the listed categories above) for the demo. I would strongly suggest using one more model to demonstrate the applicability of the developed sensors side-by-side with jGCaMP8s or iNTnC2. Since the major point of the manuscript is the new GECIs with nanomolar Ca affinities, I would recommend taking a more systematic approach and listing all known cell types and organisms with [Ca] in the lower nanomolar range and presenting it in the form of a table in the main text. It will help the reader appreciate the importance of this work more.

A1-2.

(1) To clarify the need for ultrahigh-affinity GECIs, we revised the introduction and added a supplementary item summarizing known resting [Ca²⁺]_{in} at low nM ranges in diverse organisms together with high-affinity GECIs for their detection (**Supplementary Fig. 1**).

(2) For an additional demonstration of ultra-high affinity GECIs in other biological models, we focused on cultured human cells to dissect the resting [Ca²⁺]_{in} as the transgenesis of malaria parasite or plant was challenging for us. Side-by-side analysis of GECIs in human cells demonstrated that a set of GECIs, including high-affinity variants, was useful for dissecting the resting [Ca²⁺]_{in} at the low nM range. These were added to the result part [L209-L238] and **Supplementary Fig. 9**.

Q1-3. It is unclear why the range of 20-30 nM was selected as the target. Please provide clear biological justification in the text. Again, Table with all biological systems with a known lower nM [Ca] range may help make the point here.

A1-3. Same with A1-2.

Q1-4. Further characterization of cpGINKO and cpiGLuSnFR is lacking here, it is hard to judge the significance of these sensors with the assessment of other properties, which are equally important to Kd for applicability.

A1-4. We performed a detailed analysis of the biochemical properties of iGluSnFR and GINKO by testing the effects of linker length variations (3, 6, 12, 24 amino acids) as also suggested in Q2-3.

In iGluSnFR, we found that shorter linkers yielded a higher affinity for glutamate. Different from GECIs, the optimal linker length (3 a.a.) increased the dynamic range (1.4-fold; **Supplementary Table 2**) by the dimmer and brighter intensity at the apo and saturating conditions, respectively (**Fig. 5b** and **Supplementary Table 2**).

Biochemical properties of GINKO variants were also investigated, including yet examined linker length variations (3, 6, 12 a.a.). We identified that reduced affinity of topology mutant of GINKO (cp-GINKO having a sensor-separated topology) was associated with a 1.3-fold increased signal change ($\Delta F/F_0$) at the expense of reduced brightness both at apo- and saturating conditions. Newly tested linker length variation (3, 6, 12 a.a.) in original GINKO did not increase K^+ -affinity as in the case for FRET-based K^+ indicator GEPIIs (Bischof et al., 2017), while the brightness and pK_a were not affected (**Supplementary Table 2**).

Collectively, we presented results for iGluSnFR as **revised Fig. 5** to strengthen the importance of linker length optimization in affinity tuning [L288-302]. We also presented results for GINKO as **Supplementary Fig. 12** to show the limitations of this strategy in affinity tuning [L303-L307, and L325-L332]. Detailed biochemical properties are presented in **Supplementary Table 2**.

Reviewer #2 (Remarks to the Author):

The paper focuses on developing ultrahigh-affinity single-fluorescent-protein-based GECIs for live imaging of low nanomolar range calcium dynamics. The researchers employed circular permutation to increase the affinity of existing GECIs while preserving their basic properties. The resulting GECIs, CaMPARI-nano, BGECO-nano, and RCaMP-nano, have a K_d of 20-30 nM, enabling unique biological applications such as detecting low nanomolar Ca^{2+} dynamics and multi-functional imaging with other second messengers. The study also demonstrated the effectiveness of the circular permutation strategy in modulating indicator affinity, which could be applied to other single-fluorescent-protein-based indicators of glutamate and potassium. I have the following concerns:

Q2-1. It should be noted that circular permutation of GECIs with fluorescent proteins in the circularly permuted topology is equivalent to the insertion of sensory domains into fluorescent proteins in the WT topology, which has been previously reported. While there is novelty in this paper, the topology itself is not new. The authors should acknowledge such a fact.

A2-1. We appreciate the valuable comment on the topology mutant of GECIs. We acknowledged the previous reports and modified our manuscript to focus on the linker length optimization in topology variants, which strengthened the novelty of this work. Accordingly, the manuscript has been thoroughly revised, including the title.

Title: High-affinity tuning of single fluorescent protein-type indicators by flexible linker length optimization in topology mutant.

*Q2-2. Related to the previous point, previous Ca^{2+} sensors have been developed by inserting calcium sensory domains into fluorescent proteins in the WT topology. In fact, one of the very first calcium sensors was made by inserting CaM into GFP in 1990s (PNAS 1999). The authors should review these sensors in the Introduction. It is a bit confusing that these previously reported Ca^{2+} sensors didn't show **ultralow calcium affinity**. The authors should discuss the reasons for discrepancies in their affinity compared to the **ultralow affinity achieved in this study**.*

A2-2. We agree with the importance of this comment. We revised the introduction to include camgaroo as one classical example of a topology mutant [L78-85]. We also mentioned the presence of several GECIs having a topology mutant configuration, and added a possible explanation for why NCaMP7 and NEMOs did not show ultra-high affinity in the discussion part [L315-L324].

Q2-3. The authors used 6 aa and 12 aa linkers for calcium sensors, but only used the 6-aa linker for other sensors. The reasoning behind this decision was not explained.

*A2-3. To more systematically evaluate the effects of linker length variation on the affinity, we tested 3, 6, 12 a.a., including longer ones (24 a.a.) for GECIs and others. The results revealed that each indicator has different lengths of linkers for the highest affinity, such as 12 a.a. for CaMPARI (**Supplementary Fig. 3c**), 6 a.a. for BGECO (**Supplementary Fig. 6b**)/RCaMP(**Supplementary Fig. 7b**), and 3 a.a. for iGluSnFR (**Fig. 5b,c**). We sincerely appreciate the insightful comment by the reviewer and largely revised the result and discussion part.*

*Q2-4. Further investigation is needed to understand how topology change affects the properties of the sensors. The authors could compare the initial constructs and new constructs with different linker lengths for **k(on)**, **k(off)**, **pKa**, and **thermostability**.*

*A2-4. Thanks for the valuable comment. We analyzed the effects of linker length variation on the properties of indicators for Ca^{2+} , glutamate, and K^+ . While general properties including brightness, its fold change, pK_a , Hill coefficient, and efficiency of the cellular expression, were not significantly affected in topology mutants having different linker lengths, affinities were specifically affected by linker length. To elucidate behind mechanisms, we performed kinetic analysis for GECIs. As a result, the linker length variation differently affected on- and off-kinetics in each GECI. In RCaMP, the linker elongation primarily accelerated on-kinetics (**Supplementary Fig. 7c**), as was observed for YC-nano. On the other hand, the linker elongation in CaMP2 and BGECO affected both kinetics by a slowing down of dissociation and accelerating association (**Supplementary Fig. 3d, 6c**).*

*These results were summarized in **Supplementary Table 1 and 2** and appropriately mentioned in the result part.*

For the effect of RS20 mutation and combined ones with linker length variation, please also see our reply to Q3-5.

*Q2-5. Error bars should be added to **Figure 2** and **Figure S5**, and a statistical analysis should be performed to show the superiority of CaMPARI-nano over CaMP2 (along with dead mutants and YC-Nano).*

A2-5. We added error bars to the **revised Fig. 2a** (former **Supplementary Fig 5**). In the **revised Fig. 2b** (former Fig 2), whose statistical analysis is technically difficult, we instead showed signal changes in three different ROIs supporting the reproducibility. We also compared the signal change (**Fig. 2c**), SNR (**Fig. 2d**), and decay kinetics (**Fig. 2e**) with appropriate statistical analysis, showing the superiority of developed GECIs over parental ones and jGCaMP8s ($K_d = 46\text{nM}$) in cellular Ca^{2+} imaging.

Q2-6. It would be interesting to investigate the application of this circular permutation strategy to later GCaMP variants (GCaMP6 and above).

A2-6. As commented by reviewers #1 and 3, an increased Ca^{2+} affinity in the topology mutant of GCaMP6 has been demonstrated previously (Qian et al., 2019). We appropriately cite it in the revised manuscript.

Q2-7. The fact that the cpCamP link6 sensor has almost the same K_d as the original construct suggests that linker optimization may be more important than circular permutation for increasing affinity.

A2-7. Thanks for the critical comment. Detailed analysis in this revision did reveal that the linker length optimization is the key to affinity tuning rather than circular permutation itself. Accordingly, we largely revised the result and discussion part.

Q2-8. “However, this strategy has not been considered for IFP-type GECIs because most...” and “we employed a previously unexplored strategy...”. Again, many previous sensors were based on the same topology (insertion of the sensory domain into fluorescent proteins). Other papers didn’t show that the insertion topology led to higher affinity. Would the phenomenon be largely caused by linker optimization here rather than the circular permutation itself?

A2-8. As commented by reviewers #1 and #3, an increased affinity of topology mutants has been demonstrated for GCaMP6 (Qian et al., 2019) and iGluSnFR (Wu et al., 2018). The decreased affinity of the sensor-motif separation in GINKO (this study) further supported the effectiveness of topology mutation in affinity modulation in diverse IFP-type indicators.

For the universality of the linker length optimization in topology mutants, results obtained during this revision suggest that there would be some exceptional cases as concerned by reviewer #2. The linker length variation did not increase the K^+ affinity of original GINKO, while it nicely worked for CaMP2, BGECO, RCaMP, and iGluSnFR. We have no clear explanation for such an exception, but it would be possible

that the linker length optimization in topology mutant would only work for indicators having closely positioned ends of sensing motifs in indicators having a sensor-separation topology, while additional optimization might be needed for indicators having too closely or distantly positioned sensing motif which would be investigated in future studies.

Another concern is why some indicators having the insertion topology did not show ultrahigh-affinity. The example includes camgaroo and NCaMP7/NEMOs. Different from our GECIs, camgaroo harbors a single Ca^{2+} -sensing motif (CaM) which shows one magnitude of lower Ca^{2+} affinity compared to CaM-M13. So it is reasonable for camgaroo not to show ultrahigh-affinity. Recently reported NCaMP7 and NEMO, having an insertion of CaM-RS20 in mNeonGreen, has quite similar molecular design to our GECIs. NCaMP7 and its derivative NEMOs accumulated 14 mutations on CaM, including affinity-lowering ones (e.g., N53I). We have no direct evidence, but it would be possible that the Ca^{2+} affinity of NCaMP7 and NEMO is increased by substituting mutated CaM for intact ones.

Collectively, we believe that the topology variant and linker length optimization both contribute to affinity tuning of 1FP-type indicators. To include the above discussions, we largely revised the introduction, result (**Fig. 5**), and discussion part.

Q2-9. The authors should provide more information on the sensors in Figure 5, including the dynamic change and whether the linkers were optimized for these sensors.

A2-9. We performed a comparative analysis of parental and cp-variants of iGluSnFR or GINKO for brightness, pK_a , in addition to K_d and dynamic range, confirming the major effects of linker length optimization on K_d and minor effects on other properties such as dynamic range and brightness.

Reviewer #3 (Remarks to the Author):

In this manuscript, Yusuke et al. reported a strategy that combines biosensor topology change with rationally designed sensing domain mutations to engineer high-affinity GECIs. The authors also extended the topology-altering approach to glutamate and potassium biosensors to demonstrate the applicability of this method.

Overall, I find this manuscript suitable for publication in Communications Biology. However, to better inform the readers, I believe the following concerns should be addressed.

The authors extensively used the combination of three affinity-enhancing RS20 mutations in this work. These mutations effectively decrease the GECI K_d s alongside the circular permutation strategy, yet the details of how these mutations were identified need to be included. I recommend the following changes:

Q3-1. Provide detailed information on the screening of the nine mutations, preferably with a rationale for why these particular mutations were chosen.

A3-1. Thanks for the comment. We added full information on the screening of RS20 mutation in **Supplementary Fig. 2**. Strategy was also added in the method section.

Q3-2. Specify the three key mutations in the main text when they are mentioned (e.g., Line 88, 107).

A3-2. We revised the main text to specify the three key mutations on RS20 [L105].

Q3-3. Include the site-directed mutagenesis protocol and screening protocol in the Methods section.

A3-3. Same response to Q3-1.

Q3-4. Considering the importance of these mutations, I suggest changing the title to "Affinity tuning of single fluorescent protein-type indicators by circular permutation and rational mutagenesis"

A3-4. Thanks to insightful comments from all reviewers, the major finding of our revised work turned out to be the importance of linker length optimization in the topology mutant. We thus revised the title "Affinity tuning of 1FP-type GECIs by linker length optimization in topology mutant," which would suitably strengthen our findings.

Q3-5. In the current manuscript, it is unclear to me whether the affinity change in BGECO-nano and RCaMP-nano is solely attributed to the circular permutation, the three affinity-enhancing RS20 mutations, or the combination of both strategies. Therefore, it is of utmost importance to show the calcium K_d titration curves and report the K_d values for cpB-GECO1 and cpR-CaMP1.01 (Line 129).

A3-5. Thanks for the valuable comment. K_d s of all variants of GECIs were presented in revised **Supplementary Figs. 3c, 6b, and 7b**.

To further specify the effect of linker length optimization and RS20 mutations, we analyzed k_{off} and k_{on} for all variants of GECIs (**Supplementary Table 1**), and analyzed relative changes of these kinetics to parental ones (**Supplementary Fig. 3d,e, 4c,d, 5c,d**). To our surprise, the results revealed linker length optimization and RS20 mutations differently affected on- and off-kinetics in three GECIs as follows:

We start with the most simple case for RCaMPs. **Supplementary Fig. 7c** shows how the linker length variation affected on- and off-kinetics in the absence of RS20 mutations. We observed bell shaped increase of k_{on} values peaked at 6aa linker, while k_{off} values were kept almost constant. Such a quite similar result to YC-nano (Horikawa et al., 2010) indicated that the linker length optimization specifically increased the on-kinetics in RCaMP. The effect of RS20 mutation on the non-cp variant was presented in the first two columns in **Supplementary Fig. 7d**, showing that RS20 mutation also accelerated the on-kinetics at a larger contribution ($\times 7.5$) than linker length optimization ($\times 2.5$; **Supplementary Fig. 7c**). The combination of optimized cp-linker

(6 a.a.) and RS20 mutations further accelerated on-kinetics without affecting off-kinetics (**Supplementary Fig. 7d**, red). Fold change in the increase of k_{on} was nearly $\times 10$, suggesting that optimized cp-linker (6 a.a.) and RS20 mutations additively accelerated on kinetics.

Different from RCaMP, the cp-linker optimization of BGECO affected both on- and off-kinetics with a 1.7-fold acceleration of on- and slowing down of off-kinetics ($\times 0.7$) (**Supplementary Fig. 6c**). RS20 mutations also affected both on and off kinetics with a slowing down of off- ($\times 1/2$) and acceleration of on-kinetics ($\times 1.2$) compared to parental BGECO1. A combination of cp-linker (6 a.a.) and RS20 mutations yielded no further slowing down of off-kinetics but a significant acceleration of on-kinetics than BGECO+3mut. The 4-fold increase of k_{on} was far larger than that of cp-linker ($\times 1.7$) or RS20 mutations ($\times 1.2$), suggesting that the effect of these two would be cooperative.

cp-linker optimization and RS20-mutation on CaMP2 also changed both on- and off-kinetics but in a pattern different from that of RCaMP and BGECO. **Supplementary Fig. 3d** showed that linker length optimization in topology mutant affected both on- and off-kinetics with a $1.6\times$ acceleration of on- and a $0.6\times$ slowing down of off-kinetics as observed for BGECO. RS20 mutations specifically slowed down off-kinetics ($\times 0.3$) compared to parental CaMP2, demonstrating that RS20 mutation differently affected on- and off-kinetics in three tested GECIs. A combination of cp-linker (12 a.a.) and RS20 mutations showed both further slowing down of off-kinetics and acceleration of on-kinetics than that of CaMP2+3mut. The fold change of k_{on} and k_{off} in CaMPARINano was $\times 1.4$ and $\times 0.3$ than that of parental CaMP2. These results suggested that optimized cp-linker (12 a.a.) and RS20 mutations changed on- ($\times 1.6$, and $\times 0.9 \rightarrow \times 1.4$) and off-kinetics ($\times 0.6$, and $\times 0.3 \rightarrow \times 0.2$) multiplicatively.

In summary, the linker length optimization or RS20 mutation differently affected on- and off-kinetics in three tested GECI families. The combination of these two modifications similarly increased Ca^{2+} affinity, but the behind mechanism seems to be different in each GECI such that the effects on k_{on} and k_{off} were multiplicative, cooperative, and additive, respectively. We have no clear reasoning for this, but differences in the molecular configuration of sensing modules (RS20-CaM vs CaM-RS20) or in the sterical relationship between FPs and sensing motifs (related to insertion site and linkers between them) would vary the effect of here tested modifications that would be examined in future studies.

Accordingly, we revised result part [L154-170] to include the summary of these results, and added **Supplementary Note 1** for a full discussion.

Q3-6. On Line 24 and Line 251, the authors described their strategy as a "breakthrough" and "unexplored." However, to my knowledge, GECI topology change has been reported before in the paper by Qian, Y., Rancic, V., Wu, J., Ballanyi, K., & Campbell, R. E. (2018). "A Bioluminescent Ca^{2+} Indicator Based on a Topological Variant of GCaMP6s." ChemBioChem, 20(4), 516–520. I recommend that the authors properly cite this paper and edit the wording in the current manuscript accordingly.

A3-6. We acknowledged the previous report for the topology mutant and toned down the exaggerated expression such as "breakthrough strategy".

Additional changes:

1. Authors and their contributions were updated to reflect newly performed experiments during this revision.
2. Name of GECIs were coordinated to include newly examined linker variants.
3. The identity of BGECO-nano and RCaMP-nano has been changed to accommodate 6 aa linkers, not 12 aa linkers. Accordingly, data for these two has been totally updated (Fig. 1d, 1e, Fig. 2a, Fig.4, and Supplementary Movies).
4. Because some properties of GECIs such as Kd values are highly sensitive to experimental conditions, we again performed side-by-side measurements for all indicators together with newly tested ones. Accordingly, some values were updated for slightly different ones, which do not violate our messages.
5. Methods and references were updated to reflect the above changes.
6. Typos were corrected appropriately.

Reviewers' comments:

Reviewer #1 (Remarks to the Author):

In the revised manuscript, the authors have changed the major statement, admitting that in inverted topology, linker length perhaps has a stronger influence on the affinity of the biosensors than in cpFP one. I agree with this conclusion and it is great that the authors performed additional experiments to support this conclusion, however, it also indicates that original submission was quite premature. While the authors improved the manuscript, I think they failed to address my comments and concerns in full. I provided detailed comments below.

Q1-1. I deeply regret that the authors ignored 50% of my comment. Nowhere in the rebuttal letter or the revised main text I could find a comparison of CAMPARI-nano with iNTnC2 while the authors stated that they agreed with my comment. I would like to note that authors similarly ignored comments from other Reviewers. For example, in Q2-6, the Reviewer wrote: "It would be interesting to investigate the application of this circular permutation strategy to later GCaMP variants (GCaMP6 and above)". Above here means GCaMP7 and GCaMP8 series. In the response, the authors mentioned only GCaMP6, ignoring GCaMP7 and GCaMP8. Why?

In the following comments things are getting even worse. In response to Q1-2 the authors wrote "we revised the introduction and added a supplementary item summarizing known resting $[Ca^{2+}]_i$ at low nM ranges in diverse organisms". However, it is missing in the revised manuscript. Please provide exact P and L for "supplementary item summarizing known resting $[Ca^{2+}]_i$ ". I did not find it.

Q1-3. It is unclear why the range of 20-30 nM was selected as the target. Please provide clear biological justification in the text. Again, Table with all biological systems with a known lower nM $[Ca]$ range may help make the point here. -A1-3. Same with A1-2. - I did not find the answer to Q1-3 in A1-2 - it makes me feel like the authors are making fun of me. Please provide a complete answer.

In addition, I have other comments regarding newly added data.

Figure 2c - please provide representative optical traces for CAMPARI-nano vs jGCaMP8s in dF/F scale side-by-side. Additionally, either replot Figure 2c in dF/F scale or add a new plot with dF/F (signal change in % of maximum is not canonical and quite confusing to me). Correspondingly, in the main text: "To quantitatively evaluate the utility of ultrahigh-affinity GECIs, we compared the magnitude of signal change (Fig. 2c)." - why not simply use dF/F as it is done in 99% of the publications for Ca-sensor but instead invent new parameter? Please note that in Supp Table 1 and Supp Figures 11 and 12, dF/F₀ is used, not "the magnitude of signal change". Also, the Methods section is missing a definition of "the magnitude of signal change".

Supplementary Figure 9 - single cell data should be shown, also fluorescence images for other used cell types should be provided. For the experiment shown in Supp. Fig 9, does CAMPARI-nano provide any advantages over jGCaMP8s or iNTnC2?

Supplementary Fig. 1 and Supplementary Fig. 2a have low resolution; therefore, I could not read it and review it properly. It has to be fixed before I can provide final and complete reviews of the manuscript.

In the Reporting summary checklist, the other checked box boxes for "The exact sample size (n) for each experimental group/condition, given as a discrete number and unit of measurement" and "A statement on whether measurements were taken from distinct samples or whether the same sample was measured repeatedly", however, this information is missing in Supp Figure 8, 9, 11, 12. Also, why did the authors not perform the authentication of cell lines and mycoplasma contamination tests? Are results in Supp Fig 9 potential artifacts of mycoplasma contamination?

Reviewer #2 (Remarks to the Author):

I have no further comments. The authors have appropriately considered my comments.

Reviewer #3 (Remarks to the Author):

The authors have fully addressed my concerns and comments from the previous review of this manuscript.

In the revised manuscript, the authors have changed the major statement, admitting that in inverted topology, linker length perhaps has a stronger influence on the affinity of the biosensors than in cpFP one. I agree with this conclusion and it is great that the authors performed additional experiments to support this conclusion, however, it also indicates that original submission was quite premature. While the authors improved the manuscript, I think they failed to address my comments and concerns in full. I provided detailed comments below.

[A. comment by reviewer-1]

Q1-1. I deeply regret that the authors ignored 50% of my comment. Nowhere in the rebuttal letter or the revised main text I could find a comparison of CaMPARI-nano with iNTnC2 while the authors stated that they agreed with my comment.

[our response to the comment-A]

We sincerely apologize if our actions seemed dismissive. It was not our intention to overlook the reviewers' comments. In hindsight, we acknowledge that we should have provided a clear explanation for not including the analysis for iNTnC2.

Reviewer-1 expressed concern in the summary, noting that **“moreover, several recently published GECIs exhibit Ca affinities comparable to that reported in this study while having larger fluorescence dynamic range, for example, jGCaMP8s ($K_d=46$ nM) or recent GECI iNTnC2 from Subch ($K_d=49$ nM), which should have similar performance in live cells.”**. We thus understood that comparative measurement of CaMPARI-nano with “existing high-affinity indicator” should be essential. Although reviewer-1 highlighted two examples with high Ca^{2+} -affinity, we selected jGCaMP8s for comparison. This selection was based on its higher affinity ($K_d = 46$ nM), larger dynamic range (49.5), and faster off kinetics compared to iNTnC2, since these factors are crucial for detecting subtle $[Ca^{2+}]_{in}$ changes at low nM ranges. In the first revision, we examined both the *in-vitro* and *in-cell* performance of jGCaMP8s, finding that CaMPARI-nano outperformed in most aspects, with the exception of dissociation kinetics. Had the performance of CaMPARI-nano been unsatisfactory, additional comparative measurements, including iNTnC2, would have been reasonable. However, we anticipated that jGCaMP8s would provide a more rigorous comparison. Given our findings, we concluded that a comparison with jGCaMP8s was sufficient, leading us to foregoing additional tests with iNTnC2 in the interest of resource efficiency.

In light of the possibility that iNTnC2 may outperform CaMPARI-nano and jGCaMP8s, we have conducted a comparative analysis in this revision. We began by constructing the cDNA (two codon variants optimized for human and *Dictyostelium discoideum*) of iNTnC2 with identical translated sequences as those reported by the Subach group (Korzhenovskiy. *et.al.*, Int J Mol Sci. 2022). N-

terminally poly-histidine-tagged iNTnC2 was expressed in *E.coli* and the purified proteins were subjected to biochemical analysis. The maximum brightness of mNeonGreen-based iNTnC2 was found to be 1.15-fold higher than that of mEOS2-based CaMPARIs. Our Ca^{2+} titration revealed the K_d of iNTnC2 to be 41 nM (**review-Fig.1b**), slightly lower than the previously reported 49 nM; (**review-Fig.2**). It is important to note that the dynamic range ($\Delta F/F_{\min}$) of iNTnC2, originally reported to be as high as 30 (**review-Fig.2**), could not be replicated under our experimental conditions (**review-Fig.1a**). While iNTnC2, reflecting its inversely flashing property, showed high maximum brightness under Ca^{2+} -free conditions, its minimum brightness at Ca^{2+} -saturated conditions was not low enough, leading to a significantly reduced dynamic range ($\Delta F/F_{\min} = 2.28$; **review- Fig.1a**).

Review_Fig. 1. *in vitro* property of iNTnC2. (a) Emission spectrum of purified iNTnC2. (b) Ca^{2+} -titration curve.

Figure 2. Scheme of engineering NTnC-like indicators.

Table 1. Properties of NTnC2 and iNTnC2 in vitro and in neuronal cultures. ^a Determined at 300 nM Ca^{2+} . ^b Data from [12]. ^c Data from [13]. ^d Data from [7]. ^e Data from [14].

Indicator	K_d , nM	dF/F	K_d , nM (Mg^{2+})	dF/F (Mg^{2+})	$k_{\text{on}}^{\text{obs}}$ at 300 nM Ca^{2+} , s^{-1} ^a	$t_{1/2}^{\text{off}}$, s	k_{off} , s^{-1}	dF/F Relative to RGECO1 in Neuronal Culture
R-GECO1	482 ^b	15 ^b	1138 ± 43 ^c	21.0 ± 0.2 ^c	ND	ND	0.752 ^b	1.0
GCaMP6s	144 ± 3 ^d	43 ^d	227.3 ± 0.2 ^d	46 ^d	0.49 ± 0.05 ^d	1.01 ^c	0.69 ± 0.01 ^d	3.2 ± 2 ^e
NTnC2	1408 ± 51	90	1290 ± 40	48	0.06	3.84	0.194 ± 0.001	0.95 ± 0.31
iNTnC2	49 ± 1	30	30 ± 1	10	ND	14.2	0.044 ± 0.002	0.12 ± 0.06

Review_Fig. 2. Reported properties of iNTnC2. Modified from (Korzhenevskiy. *et.al.*, Int J Mol Sci. 2022)

To further investigate whether the observed “reduced dynamic range” persisted in human and *Dictyostelium* cells, we conducted comparative *in-cell* studies with CaMPARI-nano. In HeLa cells, we evaluated the maximum and minimum brightness of iNTnC2 and CaMPARI-nano, both of which are

inversely flashing-type GECIs. When cells expressing these GECIs were treated with 1 μM of ionomycin in the presence of 1 mM extracellular Ca^{2+} concentration, the maximum intensity of CaMPARI-nano, normalized by co-expressed mCherry, reached 6.0 (**review-Fig. 3-(1)**), in contrast, iNTnC2's was notably lower at 1.4 (**review-Fig. 3-(2)**). Furthermore, after treatment with 15 μM BAPTA-AM, the minimum intensities of both CaMPARI-nano and iNTnC2 were similarly low. This led to a significantly narrower *in-cell* dynamic range for iNTnC2 (5.3; **review-Fig. 3-(3)**) compared to CaMPARI-nano (22.5; **review-Fig. 3-(4)**).

Review_Fig. 3. *in-cell* brightness of iNTnC2. Normalized intensity of HeLa cells expressing iNTnC2_P2Amcherry or CaMPARI-nano_P2Amcherry treated with BAPTA-AM (Ca^{2+} -depleted) or ionomycin (Ca^{2+} -saturation). Representative 10 cells from a single experiment for each condition. Reproducibility was confirmed by three independent experiments. ① Expected brightness of iNTnC at a similar expression/stability with CaMPARI-nano. ② Unexpected low brightness of iNTnC even at Ca^{2+} -depletion. *in-cell* dynamic range of iNTnC(③ = 5.3) and CaMPARI-nano(④ = 22.5).

To further investigate the potential temperature sensitivity of *in-cell* dynamic range of iNTnC2, Ca^{2+} imaging was performed in *Dictyostelium* cells maintained at their optimal culture temperature of 23°C. Given that ionomycin and BAPTA-AM are not suitable for *Dictyostelium* cells, we focused instead on fluorescent intensity under resting conditions. As depicted in **Figure 2a**, both CaMPARI-nano ($K_d = 19$ nM) and CaMP2_F391W+mut3 ($K_d = 43$ nM) had similarly high resting intensities when normalized by co-expressed mRFPmars. However, despite iNTnC2's comparable K_d value and *in-vitro* brightness to CaMPARIs, its resting intensity in *Dictyostelium* cells was significantly lower than CaMPARI-nano's (**review-Fig. 4**). This indicates the potential instability of iNTnC2 in *Dictyostelium* cells, similar to observations in HeLa cells. Moreover, the $[\text{Ca}^{2+}]_{\text{in}}$ transient induced by cAMP stimulation was barely detectable with iNTnC2, suggesting the *in-cell* dynamic range of iNTnC2 was significantly lower than that of CaMPARI-nano and jGCaMP8s (**review-Fig. 4**). Importantly, physiological $[\text{Ca}^{2+}]_{\text{in}}$ transients associated with spontaneously synthesized cAMP remained undetectable with iNTnC2 (**review-Fig. 5**).

Review_Fig. 4. Performance of iNTnC2 in cAMP stimulated *D.discoideum* cells. Normalized intensity of *D.discoideum* cells expressing CaMPARI-nano, iNTnC2, and jGCaMP8s. Mean \pm SEM. N = 5 cells. Reproducibility was confirmed by three independent experiments. ① Expected brightness of iNTnC2 for similar expression/stability with CaMP2-nano. ② Low resting brightness of iNTnC2 than that of CaMP2-nano. ③ Significantly smaller dynamic range of iNTnC2.

Review_Fig. 5. $[Ca^{2+}]$ dynamics for spontaneously signalling *D.discoideum* cells. Normalized intensity of *D.discoideum* cells expressing CaMPARI-nano, iNTnC2, and jGCaMP8s. Traces of their representative ROIs each containing ~ 10 cells.

In summary:

Despite their high Ca^{2+} -affinities, both iNTnC2 and jGCaMP8s did not match the performance of CaMPARI-nano in living cells, especially regarding spontaneous $[Ca^{2+}]_{in}$ transients in *Dictyostelium* cells. iNTnC2 showed limited effectiveness in both HeLa and *Dictyostelium* cells, attributed to its poor *in-cell* stability and a dynamic range that was narrower than anticipated, even under *in-vitro* conditions. While these findings are considered for inclusion in the manuscript, our goal is not to discredit iNTnC2 but to highlight areas for further investigation, given its observed dynamic range.

To clarify our methodology, we have provided a concise explanation for selecting jGCaMP8s for comparative studies. This is outlined as follows:

L204 (result):

“We also included the recently developed jGCaMP8s as a representative of existing GECIs with high Ca^{2+} -affinity and rapid kinetics⁷.”

[B. comment by reviewer-1]

I would like to note that authors similarly ignored comments from other Reviewers. For example, in Q2-6, the Reviewer wrote: “It would be interesting to investigate the application of this circular permutation strategy to later GCaMP variants (**GCaMP6 and above**)”. Above here means GCaMP7 and GCaMP8 series. In the response, the authors mentioned only GCaMP6, ignoring GCaMP7 and GCaMP8. Why?

[our response to comment-B]

We recognize the potential confusion caused by the phrase “6 and above”, which in a mathematical context, typically indicates inclusivity. Furthermore, we interpreted the question in Q2-6 as highlighting a point of interest from the reviewer-2 rather than a mandatory inquiry, promoting us to respond within our experimental limitations. Although reviewer-2 did not raise any concerns on this issue, we understand the importance of clarity in our responses for explicitly stating like that “We have not tested jGCaMP7 and 8. However, their similar molecular configuration to GCaMP6 and GECOs leads us to hypothesize that circularly permuted variants of jGCaMP7 and 8 would also exhibit increased Ca^{2+} -affinity as demonstrated with GCaMP6.”

To address this oversight, we further explored the effect of circular permutation on the most recent jGCaMP8 variant. This involved integrating a 20 amino acid sequence from ENOSP as a binding target for CaM- Ca^{2+} , instead of the conventional 21 amino acid sequence from RS20. From the three jGCaMP8 variants (sensitive, moderate, and fast), we selected jGCaMP8f for its lower Ca^{2+} affinity, which allows for Ca^{2+} titration across a broader concentration range. We constructed a circularly permuted variant of jGCaMP8f (cp_jG8F), employing a long and flexible linker ([GGTGGGS]_{x4}), as shown in (**review-Fig.6a**). As expected, cp_jG8F exhibited a higher Ca^{2+} affinity ($K_d = 175$ nM) compared to its parental form (230 nM, as measured; **review-Fig.6b**). Despite observing a discrepancy in the reported K_d value of parental jGCaMP8f (330 nM in ref7), which may be due to variations in experimental setups, we maintain confident in our Ca^{2+} titration method, ensured by rigorous cross-calibration using Quin2 ($K_d = 62$ nM) and Fluo4 ($K_d = 345$ nM).

Review-Fig.6. Increased Ca^{2+} -affinity of the latest version of GCaMP. (a) Molecular design of jGCaMP8f and circularly permuted one incorporating long and flexible linker. (b) Ca^{2+} titration curve for purified jGCaMPf proteins.

In conclusion, circular permutation was found to be effective in enhancing the Ca^{2+} -affinity of the latest version of jGCaMP8f. Although we have not extended our testing to jGCaMP7 or other GECIs such as XCaMPs, we anticipate that similar enhancements in Ca^{2+} -affinity might be observed in these variants as well.

While we hope that our results address the reviewers' interest, we believe that incorporating these results into the current manuscript may not directly contribute to its central thesis. Thus, we have not chosen to include these specific results and discussion in this revision but remain open to further exploration in future work.

[C. comment by reviewer-1]

In the following comments things are getting even worse. In response to Q1-2 the authors wrote "we revised the introduction and added a supplementary item summarizing known resting $[\text{Ca}^{2+}]_{\text{in}}$ at low nM ranges in diverse organisms". However, it is missing in the revised manuscript. Please provide exact P and L for "supplementary item summarizing known resting $[\text{Ca}^{2+}]_{\text{in}}$ ". I did not find it.

[our response to comment-C]

We apologize for any confusion caused by the low resolution of **Supplementary Fig 1.**, and our failure to clearly highlight the revised sections in the maintext.

Below, we present the section of the main text that should have been clearly marked in red in the previous version of manuscript:

L.59-64 (introduction): modified in this second revision was highlighted in red.

“However, they are not suitable for detecting $[Ca^{2+}]_{in}$ dynamics in certain living samples with low nanomolar levels (<100 nM) of resting $[Ca^{2+}]_{in}$ and amplitude of $[Ca^{2+}]_{in}$ transients, such as those found in malaria parasites³, plant cells⁴, murine astro-glia⁵, and slim molds⁶ (**Supplementary Fig. 1**). To monitor $[Ca^{2+}]_{in}$ dynamics at low nanomolar levels, ultrahigh-affinity indicators whose K_d is in the middle of the $[Ca^{2+}]_{in}$ ranges are required (**Supplementary Fig. 1**).”

Unedited high-resolution **Supplementary Fig1** is as follows:

estimated resting $[Ca^{2+}]_{in}$ (nM)	organism	celltype	GECI	GECI's K_d (nM)	references
10-15	D. discoideum	developing cell	YC-Nano15	15	Ref(6)
23-37	P. falciparum	trophozoite stage	Campari-nano	19	This work
32-59	R. norvegicus	pyramidal neuron	YC-nano50	48.5	Ref(3)
40-50	A. thaliana	root cell	OGB1	206	Ref(16)
40-50	R. norvegicus	astro-glia	YC-Nano65	65	Ref(4)
40-90	R. norvegicus	astro-glia	OGB2 (FLIM)	158	Ref(15)
40-90	M.musculus	astro-glia	YC-Nano50	48.5	Ref(5)

Supplementary Fig. 1. Known resting $[Ca^{2+}]_{in}$ at the low nM range.

Representative of resting $[Ca^{2+}]_{in}$ at low nM in various celltypes and organisms. Sorted by resting $[Ca^{2+}]_{in}$.

[D. comment by reviewer-1]

Q1-3. It is unclear why the range of 20-30 nM was selected as the target. Please provide clear biological justification in the text. Again, Table with all biological systems with a known lower nM [Ca] range may help make the point here. -A1-3. Same with A1-2. – I did not find the answer to Q1-3 in A1-2 – it makes me feel like the authors are making fun of me. Please provide a complete answer.

(1) Please provide clear biological justification in the text (why the range of 20-30 nM was selected as the target);

[our response to comment-D1]

We would like to clarification on the specific biological justifications requested in this context. Successful Ca^{2+} imaging should utilize GECIs whose K_d value aligns closely the midpoint between resting and peak responses. GECIs with lower affinity are often used in conventional Ca^{2+} imaging, as they detect Ca^{2+} transient with high $\Delta F/F_0$ and SNR. However, this approach often does not consider the potential difference in resting levels among cells, a factor whose importance for controlling response amplitude magnitude has recently been recognized (ref15 in the main text). For a specific example, in *Dictyostelium* cells, the resting and peak $[Ca^{2+}]_{in}$ for spontaneous activities are estimated at 10 and 30 nM, respectively. Therefore, based on our analyses, the ideal K_d value for

GECIs in our study is determined to be around 20 nM, a conclusion that is thoroughly supported by the data presented in this work.

For a detailed analysis of low resting Ca^{2+} concentration, ultra-high affinity GECIs prove to be more crucial.

As detailed in **Supplementary Fig.1**, various cell types across multiple organisms have been shown to have resting Ca^{2+} levels around 30 to 40 nM. Thus, we set our target affinity range to 15-30 nM. Accordingly, we revised the two parts of the main text as follows:

L.59-64 (introduction):

“However, they are not suitable for detecting $[\text{Ca}^{2+}]_{\text{in}}$ dynamics in certain living samples with low nanomolar levels (<100 nM) of resting $[\text{Ca}^{2+}]_{\text{in}}$ and amplitude of $[\text{Ca}^{2+}]_{\text{in}}$ transients, such as those found in malaria parasites³, plant cells⁴, murine astro-glia⁵, and slim molds⁶ (**Supplementary Fig. 1**). To monitor $[\text{Ca}^{2+}]_{\text{in}}$ dynamics at low nanomolar levels, ultrahigh-affinity indicators whose K_d is in the middle of the $[\text{Ca}^{2+}]_{\text{in}}$ ranges are required (**Supplementary Fig. 1**).”

L.100 (result):

“To develop 1FP-type GECIs with ultrahigh-affinity suitable for detecting $[\text{Ca}^{2+}]_{\text{in}}$ at the low nM range (15–30 nM, covering the lowest resting $[\text{Ca}^{2+}]_{\text{in}}$ in *Dictyostelium discoideum* cells; **Supplementary Fig. 1**), we aimed to enhance the Ca^{2+} affinity of CaMPARI2¹⁰.”

(2) **A table with all biological systems with a known lower nM [Ca] range may help make the point here.**

[our response to comment-D2]

Please find **Supplementary Fig.1** at a high resolution in [our response to comment-C].

In addition, I have other comments regarding newly added data.

[E. comment by reviewer-1]

Figure 2c – please provide representative optical traces for CAMPARI-nano vs jGCaMP8s in dF/F scale side-by-side.

[our response to comment-E]

Right (**review-Fig. 7**) is the representative optical traces for CAMPARI-nano vs jGCaMP8s in dF/F scale side-by-side (continued to the next part).

Review_Fig.7. A side-by-side representation of optical traces for CAMPARI-nano vs jGCaMP8s in normalized intensity (left) or deltaF/F_resting (right) scale.

[F. comment by reviewer-1]

Additionally, either replot Figure 2c in dF/F scale or add a new plot with dF/F (signal change in % of maximum is not canonical and quite confusing to me).

[our response to comment-F]

We prepared a series of replotting figures for **Fig. 2c**, now shown as **review-Fig.8**, in response to the reviewer's comment that requested clarification on the denominator F used in our calculations. **Review-Fig.8a** presents ΔF without normalization, offering a straightforward comparison of changes in fluorescence. Thus, comparisons among indicators are only applicable to groups that share imaging conditions (*i.e.*, the green, blue, and red groups). **Review-Fig.8b** and **8c** show ΔF normalized by either F_{resting} or F_{min} , respectively. In **review-Fig.8b**, $\Delta F/F_{\text{resting}}$ exhibits comparable magnitudes for both CaMPARI-nano and jRCaMP8s, consistent with observations in **review-Fig.7**. We agree that the normalization of ΔF with F_{resting} is widely utilized in a variety of studies. However, this applies only to comparative analysis for identical types of GECIs, both flashing-type and inversely flashing GECIs. For example, when there are two different types of GECIs with identical K_d and dynamic range, it is expected to observe cellular $[Ca^{2+}]_{\text{in}}$ transient with identical ΔF for these two. However, the resting intensities intrinsically differ, being lower for flashing GECIs and higher for inversely flashing ones. Thus, normalization by resting intensity becomes less effective for comparing the performance between flashing and inversely flashing GECIs due to the underestimation of performance for the latter caused by the larger denominator. An exceptionally low resting brightness in flashing type GECIs (0.001) can be another risk causing overestimation of GECIs' performance having a low maximum brightness (0.1), yielding a high $\Delta F/F_{\text{resting}}$ of 100 caused by the small denominator.

F_{min} might serve as an alternative denominator for normalization; however, it varies in a context-dependent manner and is not suitable for our purpose. (continued to the next part)

Review_Fig. 8.

Replot of Figure 2c in un-normalized ΔF (a), and normalized ΔF with resting F (b), minimum F (c), and maximum ΔF specific to each indicator (d).

Schematic representation of denominator F(0) utilized for each normalization (right).

[G. comment by reviewer-1]

Correspondingly, in the main text: "To quantitatively evaluate the utility of ultrahigh-affinity

GECIs, we compared the magnitude of signal change (Fig. 2c).” – why not simply use dF/F as it is done in 99% of the publications for Ca-sensor but instead invent new parameter? Please note that in Supp Table 1 and Supp Figures 11 and 12, dF/F_0 is used, not “the magnitude of signal change”. Also, the Methods section is missing a definition of “the magnitude of signal change”.

(1) why not simply use dF/F as it is done in 99% of the publications for Ca-sensor but instead invent new parameter?

[our response to comment-G1]

The purpose of our study, as shown in **main-Fig.2c**, is to compare the performance of a variety of GECIs, including both different types (flashing or inversely flashing) and various imaging conditions (green, blue, and red). For this purpose, we chose to normalize ΔF against the maximum range of the signal changes as the most appropriate method. For example, CAMPARI-nano reports a spontaneous Ca^{2+} change with a larger brightness change corresponding to 40% of its maximal signal change, whereas jGCaMP8s demonstrates a 5% intensity change at its maximum capacity, while CAMPARI-nano and jGCaMP8s share the maximum range of signal changes (**Supplementary Fig.8a**).

(2) Please note that in Supp Table 1 and Supp Figures 11 and 12, dF/F_0 is used, not “the magnitude of signal change”.

[our response to comment-G2]

In **Supplementary Table 1**, we apply the formula $\Delta F/F_0$ ($[F_{\max}-F_{\min}]/F_{\min}$) for *in vitro* studies because F_{\max} and F_{\min} values can be precisely determined,, independent of varying experimental conditions.

In **Supplementary Figures 11 and 12**, we employed $\Delta F/F_0$ as these exclusively evaluate the performance of flash-type indicators.

(3) the Methods section is missing a definition of “the magnitude of signal change”.

[our response to comment-G3]

In the previous manuscript, the definition of “the magnitude of signal change” was visually presented in **Supplementary Fig. 8**. and described in its legend. Also we had included a navigation message in the legend of **Fig 2**; however, we realized that the statement “See Supplementary Fig. 8a for a quantification method.” was incorrectly referenced to Fig 2b instead of Fig 2c. Therefore, we have corrected this oversight by appropriately referencing Supplementary Fig. 8a in the legend of Fig. 2c and by enhancing the navigation within the Method section.

Collectively, we greatly appreciate the comments provided, as they highlight a question likely shared by readers from various fields: why we did not utilize $\Delta F/F_0$. To help readers’ understanding, we

revised the main text and legend of Fig2c as follows:

L.200-206 (result):

“Specifically, to enable simultaneous comparison of the signal change magnitude between both flashing and inversely flashing GECIs of different colors, we measured the fractional change of given indicators to their maximal response (see **Supplementary Fig. 8** for a more detailed definition), since the commonly utilized $\Delta F/F_{resting}$ is not suitable for comparing the performance of flashing and inversely flashing GECIs.”

L.613-616 (legend of Fig2c):

“(c, d) The magnitude of signal change, defined as the ratio of a **given response to the maximum signal change in each GECI** (c), and the peak signal-to-noise-ratio (SNR, d) for spontaneous $[Ca^{2+}]_{in}$. See **Supplementary Fig. 8a** for the quantification method used in Fig. 2c.”

We also added a navigation statement in the methods section as follows.

L.474-475 (Method):

“The calculation of the magnitude of signal change in Fig. 2c is detailed visually in Supplementary Fig.8 and described in its legend.”

[H. comment by reviewer-1]

Supplementary Figure 9 – single cell data should be shown, also fluorescence images for other used cell types should be provided. For the experiment shown in Supp. Fig 9, does CAMPARI-nano provide any advantages over jGCaMP8s or iNTnC2?

(1) Supplementary Figure 9 – single cell data should be shown, also fluorescence images for other used cell types should be provided.

[our response to comment-H1]

Supplementary Figure 9 was updated to show the single-cell data (**9b** and **9c**) and fluorescence images for all celltypes as below.

Supplementary Fig. 9. Estimation of resting $[\text{Ca}^{2+}]_{\text{in}}$ among diverse celltypes.

(a) The cytoplasmic expression of GECIs, reference mCherry, and ratio image of GECI/reference in HEK293, HeLa, U-87 cells at 48 h after the transfection. Those of astro-glia at 4 days after transfection. Scale bar: 0.05 mm. (b) Resting $[\text{Ca}^{2+}]_{\text{in}}$ detected by affinity variants of CaMPARIs. The intensity ratio in resting HEK293, HeLa, and U-87 cells ($N = 10$ cells) was shown. The range of GECI's maximum response was determined by the intensity ratio in BAPTA-AM or ionomycin-treated cells. (c) Boxplot of the fractional changes of each GECIs. The range of GECI's maximum response in astro-glia ($N = 9$ cells) was analyzed by using independently determined R_{max} (4.1) and R_{min} (0.1), since its culture condition (4 days after transfection) was different from the other three celltypes (48 h after transfection). Median, interquartile points, and data range were shown, respectively. Shown are the data in a representative single experiment whose reproducibility was confirmed by three independent experiments.

(2) For the experiment shown in Supp. Fig 9, does CaMPARI-nano provide any advantages over jGCaMP8s or iNTnC2?

[our response to comment-H2]

We are concerned that we may not have fully grasped the rationale behind this comment. However, a concise response would be that CaMPARI-nano does not hold a distinct advantage over jGCaMP8s due to the mismatch between the K_d of ultrahigh-affinity CaMPARI-nano and the $[Ca^{2+}]_{in}$ levels observed in cultured mammalian cells.

Nevertheless, as outlined in our manuscript (**Line.223-226, and Line.248-250**), our experiments were designed to demonstrate the utility of “a set of GECl including high-affinity variants” in assessing varying $[Ca^{2+}]_{in}$ level in cultured cells.

In this context, we would like to emphasize that a set of affinity variants, including CaMPARI-nano, has advantages over jGCaMP8s.

For a balanced discussion, we suggest starting with a comparison with the performance between CaMP2_F391+mut3 and jGCaMP8s, both sharing a K_d value of ~40 nM. In cultured cells, as illustrated in **review-Fig.9**, jGCaMP8s exhibited a typical cellular distribution with resting cells displaying a low intensity ratio (around 1.0), aligning with its flashing property. When the fractional response was analyzed by

Review_Fig. 9. performance of jGCaMP8s in mammalian cells.

(a) The cytoplasmic expression of GECl, reference mCherry, and ratio image of GECI/reference in HEK293, HeLa, U-87 cells at 48 h after the transfection. **(b)** Comparative measurement of “maximum range of signal change” for CaMPARIs and jGCaMPs. Notice that 1/3 reduced the maximum intensity ratio of jGCaMP8s compared to CaMPARIs, while the DR of jGCaMP8s was preserved to be high ($\Delta R/R_{min} = 27.9$). **(c)** The fractional response of CaMP2_F391W+3mut (left) and jGCaMP8s (right) both have $K_d = 40$ nM. Fractional change determined by bright CaMP2_F391W+3mut was basically reproduced by jGCaMP8s.

measuring intensity changes following BAPTA-AM and ionomycin-treatment, jGCaMP8s exhibited a pattern of fractional change in cultured cells that was closely comparable to that of CaMP2_F391W+3mut (**Supplementary Fig. 9 and review-Fig.9c**). Therefore, it is evident that both jGCaMP8s and CaMP2_F391W+3mut possess a comparable proficiency in delineating the resting

$[Ca^{2+}]_{in}$ levels in cultured cells.

In the following discussion, we aim to compare a single GECI to a set of GECI variants, where the latter demonstrates advantages over the former. Estimating resting $[Ca^{2+}]_{in}$ levels using a single GECI such as jGCaMP8s requires the concurrent use of BAPTA-AM and ionomycin-treatment. Conversely, a set of GECI variants spanning low, moderate, and ultra-high affinities enables the rapid and straightforward estimation of resting $[Ca^{2+}]_{in}$ levels from their snapshots, even without BAPTA-AM and ionomycin-treatments. Although mammalian cells are highly sensitive to these chemicals, it is well-known that certain cell types and organisms display insensitivity to BAPTA-AM and/or ionomycin. Examples include plant cells (PMID: 33863225) and *Dictyostelium* cells (PMID: 2854083.), which are atypical yet extensively studied model organisms. These organisms exhibit resistance to certain chemicals due to extracellular esterase activity or unique membrane properties. In these models, accurately estimating resting $[Ca^{2+}]$ levels have been a significant challenge. However, a set of CaMPARI variants enables analyses that are not possible with jGCaMP8s, offering a more versatile approach to understanding $[Ca^{2+}]$ dynamics.

In addition to the conceptual benefits aforementioned, CaMPARIs also exhibit practical advantages over jGCaMP8s. As demonstrated in **review-Fig.9b**, the maximum brightness of jGCaMPs was found to be lower compared to that of CaMPARIs. The intensity ratio of jGCaMP8s in ionomycin-treated HeLa cells was 2.02. This is approximately one-third of that observed for CaMP2_F391W+mut3, despite both exhibiting a significant *in-cell* dynamic range of approximately 30.

Considering the similarly high maximum brightness of jGCaMP8s and CaMP2_F391W+mut3 as shown in **Supplementary Table 1**, it is expected that both indicators would exhibit comparable intensity ratios. Therefore, based on our observations, jGCaMP8s appears to exhibit reduced stability in mammalian cells in comparison to CaMPARIs, as indicated by the variability in intensity ratio. The unexpectedly low fluorescence signals from jGCaMP8s may hinder the accurate detection of resting $[Ca^{2+}]_{in}$ levels.

Upon detailed examination of data for jGCaMP8s (**review-Fig.9c**), both GECIs effectively detected significant difference in resting $[Ca^{2+}]$ levels between HEK293 and HeLa versus U-87 and astro-glia. However, the subtle differences between HEK293 versus HeLa and U-87 versus astroglia cells, identified by CaMP2_F391W+mut3, cannot be clearly discerned using jGCaMP8s. Based on the observed data, it is plausible to infer that CaMPARIs offer superior detection sensitivity compared to jGCaMP8s, likely due to their higher stability in mammalian cells.

While CaMPARI-nano does not outperform existing high-affinity indicators in direct comparisons, a comprehensive range of CaMPARI variants, spanning low to ultra-high affinity, offers significant

conceptual and practical benefits, such as broader applicability across various cellular contexts. These results and discussions enhance the depth of our manuscript, although they are not the cornerstone of its theoretical framework. Additionally, the observed reduced brightness of jGCaMP8s, although not a primary focus of our current study, highlights an area for detailed investigation in future research.

We have therefore updated **Supplementary Fig.9** to include data for astro-glia by using a newly obtained data set in this revision. To ensure the accuracy of this sensitive analysis, all experiments were conducted under identical experimental conditions, including culture, transfection, and imaging. Also, updated experimental conditions yielded slightly different values for intensity ratios and fractional changes from those previously presented, yet they still preserved the relative relationship of resting $[Ca^{2+}]_{in}$ among celltypes at a fine resolution. To accommodate these changes, the main text was also revised as follows:

L.223-250 (result part in maintext):

“Attributing inconsistent estimation of resting $[Ca^{2+}]_{in}$ to insufficient detection sensitivity at a low nM range, we evaluated the relative levels across different cell types. This was achieved by comparing the fluorescence intensities of CaMPARI2 variants, which span low-to-ultrahigh Ca^{2+} affinities. CaMPARI2 variants, when co-expressed with the reference marker P2A-mCherry, exhibited uniform cellular distribution as shown in **Supplementary Figure 9a**. This observation indicates that RS20-mutations and topology mutation do not adversely affect the expression of these indicators. In HEK293 cells, treatment with BAPTA-AM or ionomycin resulted in CaMPARI2 fluorescence ratio (R : GECIs/reference) of 6.17 for Ca^{2+} depletion and 0.20 for Ca^{2+} saturation. This provided a substantial dynamic range of 5.97, as detailed in **Supplementary Fig. 9b**. In non-treated HEK293 cells, CaMP2_F391W (*in vitro* $K_d = 121$ nM) and CaMPARI-nano (*in vitro* $K_d = 19$ nM) showed fluorescence ratio of 5.4 and 0.25, respectively, as shown in **Supplementary Fig. 9b**. This resulted in fractional changes for each indicator of 13 and 99%, respectively, detailed in **Supplementary Fig. 9c**. The fractional change for CaMP2_F391W+3mut was 64% ($R = 2.3$; **Supplementary Fig. 9c**), indicating that the resting $[Ca^{2+}]_{in}$ in HEK293 cell is approximately 50 nM, slightly above the *in vitro* K_d value of 40 nM for CaMP2_F391W+3mut. Similar results were obtained for HeLa cells with smaller fractional changes for three variants of CaMPARI2 (**Supplementary Fig. 9c**), indicating that HeLa cells have a slightly lower resting $[Ca^{2+}]_{in}$ than HEK293 cells. Glioma-derived U-87 cells exhibited even lower resting $[Ca^{2+}]_{in}$ levels, with CaMP2_F391W+3mut and CaMPARI-nano showing fractional changes of 23% and 94%, respectively (R values of 4.8 and 0.32, as detailed in **Supplementary Fig. 9b, c**). Thus, the resting $[Ca^{2+}]_{in}$ in U-87 cells was estimated to be approximately 30 nM, which lies between the K_d values of 43 nM and 19 nM. Primary cultures of astro-glia showed a very similar pattern of fractional change as observed in U-87 cells (**Supplementary Fig. 9c**), suggesting that a low resting $[Ca^{2+}]_{in}$ is likely a common characteristic among glia-related cells. While

resting $[Ca^{2+}]_{in}$ levels can be determined more precisely using sophisticated quantification methods^{17, 19, 22}, a set of GECl, including high-affinity variants, offers practical approach for rapid estimation and comparison of low nM range $[Ca^{2+}]_{in}$ levels across various cell types.”

The revised version of **Supplementary Fig. 9** was shown in [our response to comment-H1].

[I. comment by reviewer-1]

Supplementary Fig. 1 and Supplementary Fig. 2a have low resolution; therefore, I could not read it and review it properly. It has to be fixed before I can provide final and complete reviews of the manuscript.

[our response to comment-I]

Supplementary Fig.1 at high resolution was presented in [our response to comment-C].

Below is the **Supplementary Fig.2a** at high resolution.

If the reviewer finds the resolution unsatisfactory, please immediately request a high-resolution one from the editor.

b

mutations on RS20	K_d (approx.)
L398	121 nM
R389V	100 nM
L398I	90 nM
G395A	75 nM
G395A, R389V	75 nM
G395A, A400L	75 nM
G395A, L398I	70 nM
G395A, H396W	60 nM
H396W	65 nM
H396W, L391I	55 nM
G395A, H396W, L398I	45 nM

Supplementary Fig. 2. High-affinity mutations in RS20 of CaMP2_F391W.

Screening of high-affinity mutations in RS20 of CaMP2_F391W. Gray represents tested mutations yielding preserved- or lower affinity (low). Colors discriminate the magnitude of increased affinity, where magenta shows the higher affinity. **(b)** Effects of combinations of selected mutations in (a). We first focused on G395A since it is a known mutation increasing affinity in CaMPARI2¹⁰. The combination of G395A with R389V or A400L did not increase Ca^{2+} affinity, and R399M was found to reduce the dynamic range. On the other hand, L398I and H396W were found to increase Ca^{2+} affinity when combined with G395A. We finally tested the triple mutation and found that G395A, H396W, and L398I further increased Ca^{2+} affinity than those in any of the two without affecting other properties, such as dynamic range, brightness, and photoconvertibility.

[J. comment by reviewer-1]

In the Reporting summary checklist, the other checked box boxes for “The exact sample size (n) for each experimental group/condition, given as a discrete number and unit of measurement’ and “A statement on whether measurements were taken from distinct samples or whether the same sample was measured repeatedly”, however, this information is missing in Supp Figure 8, 9, 11, 12.

[our response to comment-J]

Thanks for the detailed check in the Reporting summary checklist. Description in Supp Figures 8, 9, 11, 12. were corrected appropriately.

L97 in Supplementary Information (**Supp. Fig. 8**, legend)

Shown are the data in a representative single experiment whose reproducibility was confirmed by three and two independent experiments for (a) and (b), respectively.

L111 in Supplementary Information (**Supp. Fig. 9**, legend)

Shown are the data in a representative single experiment whose reproducibility was confirmed by three independent experiments.

L129 in Supplementary Information (**Supp. Fig. 11**, legend)

Shown are the data in a representative single experiment whose reproducibility was confirmed by three independent experiments.

L138 in Supplementary Information (**Supp. Fig. 11**, legend)

Shown are the data in a representative single experiment whose reproducibility was confirmed by two independent experiments.

[K. comment by reviewer-1]

Also, why did the authors not perform the authentication of cell lines and mycoplasma contamination tests?

[our response to comment-K]

Thank you for providing this clarification regarding the authentication of cell lines and mycoplasma contamination tests. While these tests were not directly performed on the cells used in our study, it is important to note that the cells used were at low passages (2 for HEK293 and Hela, and 5 for U-87 cells) that had previously passed these tests at the original cell banks (RIKEN and ATCC). The decision not to directly conduct these tests was due to the associated costs and time required for third-party services. However, it's worth noting that cells at low passages are generally considered less prone to cross-contaminate. To address potential mycoplasma contamination, we conducted a PCR check and

confirmed that all cell lines used in this study are free from mycoplasma (review-Fig.10).

Accordingly, we revised the Reporting summary checklist as follows.

Review_Fig. 10. PCR test for mycoplasma contaminations. HEK293 (1), HeLa (2), U-87 (3), Ax2 (4), and kit-supplied positive control (5, 750-bp). M is a 1kb-plus DNA ladder.

[Eukaryotic cell lines]

Cells at a low passage (2 for HEK293 and HeLa, and 5 for U-87 cells), which have been authenticated at original cell banks (RIKEN and ATCC), were utilized in this study.

[Mycoplasma contamination]

Free of mycoplasma contamination has been confirmed by the PCR check using TaKaRa PCR Mycoplasma Detection Set.

Eukaryotic cell lines	
Policy information about cell lines and Sex and Gender in Research	
Cell line source(s)	Ax2 derived from dicty stock center. HeLa and HEK293 from RIKEN BRC. U-87 from ATCC.
Authentication	Cells at a low passage (2 for HEK293 and HeLa, and 5 for U-87 cells), which have been authenticated at original cell banks, were utilized in this study.
Mycoplasma contamination	
Commonly misidentified lines (See ICLAC register)	Free of mycoplasma contamination has been confirmed by the PCR check using TaKaRa PCR Mycoplasma Detection Set
	NA

[L. comment by reviewer-1]

Are results in Supp Fig 9 potential artifacts of mycoplasma contamination?

[our response to comment-L]

We have confirmed that U-87 cells used in our experiments are free from mycoplasma contamination. To validate the low resting Ca^{2+} concentrations observed in glioma-derived U-87 cells, we also assessed resting Ca^{2+} levels in primary astro-glia as shown in **Supplementary Figure 9**. The results showed that primary astro-glia exhibited similarly low resting $[\text{Ca}^{2+}]_{\text{in}}$ levels as observed in U-87 cells (indicated by the red circle in the right Figure, suggesting this could be a characteristic feature of glia-related cells.

Revised Supplementary Fig. 9c.

Accordingly, we have updated **Supplementary Fig. 9c** in line with our response to comment-H1.

The main text has been revised as follows:

L.245-247 (result part in maintext):

“Primary cultures of astro-glia demonstrated a very similar pattern of fractional change as observed in U-87 cells (**Supplementary Fig. 9c**), suggesting that a low resting $[\text{Ca}^{2+}]_{\text{in}}$ is likely shared among glia-related cells.”

Procedures for preparation, transfection, and imaging of astro-glia were also updated in the method section.

L.436-444 (methods part in maintext):

“Primary cultures of astro-glia were prepared by differentiation of neurosphere prepared from the mouse ganglionic eminence at embryonic day 18.5, as described previously³⁷. Briefly, neurospheres were dissociated into single cells and plated onto 35-mm glass bottom dishes. These cells were cultured in Neurobasal medium supplemented with LIF and BMP-2 for astrocytes differentiation. One day after differentiation, cells were transfected with GECIs expression vectors using Lipofectamin 2000 (Thermo Fisher Scientific). Imaging was conducted at four days after DNA transfection. All experimental procedures using mice were performed in accordance with the ethical guidelines of Tokushima University, and this study was approved by the Animal Research Committee, Tokushima University.”

REVIEWERS' COMMENTS:

Reviewer #1 (Remarks to the Author):

I really appreciate the Authors' thorough and meticulous response to all my comments (and I also apologize for taking my time to review the response and failing to provide my comments in time due to unforeseen obstacles in my daily academic routine activities). I also acknowledge the Authors' hard work in carrying out all additional experiments to demonstrate the superiority of CaMPARInano over other Ca-sensors with similar Kd under the tested conditions. I agree with all conclusions and the rationale behind the experimental results. The authors also clearly explained the metrics used for sensor comparison. Overall, the Author addressed all my concerns in full, and I do not have any further comments. However, I wonder why the new experimental results were not included in the final version of the manuscript; these results clearly demonstrate the significance of this study and the applicability of the developed sensors. Moreover, these results can help end users to choose the right sensors for a given application. As often, end users choose sensors based on the literature data from independent articles without direct comparison under matching conditions. Here, the Author performed a careful side-by-side comparison. I believe it will be helpful for end-users in Ca-sensor selection. New data can be simply added at Supp Figures with brief mentions in the main text (please consider this possibility).

We greatly appreciate reviewer#1's efforts in reviewing. It is a pleasure to hear his/her positive comments, such as “*the Author addressed all my concerns in full, and I do not have any further comments.*”

REVIEWERS' COMMENTS:

Reviewer #1 (Remarks to the Author):

I really appreciate the Authors' thorough and meticulous response to all my comments (and I also apologize for taking my time to review the response and failing to provide my comments in time due to unforeseen obstacles in my daily academic routine activities). I also acknowledge the Authors' hard work in carrying out all additional experiments to demonstrate the superiority of CaMPARInano over other Ca-sensors with similar Kd under the tested conditions. I agree with all conclusions and the rationale behind the experimental results. The authors also clearly explained the metrics used for sensor comparison. Overall, the Author addressed all my concerns in full, and I do not have any further comments. However, I wonder why the new experimental results were not included in the final version of the manuscript; these results clearly demonstrate the significance of this study and the applicability of the developed sensors. Moreover, these results can help end users to choose the right sensors for a given application. As often, end users choose sensors based on the literature data from independent articles without direct comparison under matching conditions. Here, the Author performed a careful side-by-side comparison. I believe it will be helpful for end-users in Ca-sensor selection. New data can be simply added at Supp Figures with brief mentions in the main text (please consider this possibility).

The suggestion by the reviewer:

However, I wonder why the new experimental results were not included in the final version of the manuscript; these results clearly demonstrate the significance of this study and the applicability of the developed sensors. Moreover, these results can help end users to choose the right sensors for a given application.

Our response to this suggestion:

Thanks for this comment. According to reviewer-1's suggestion, we included data showing unexpectedly attenuated performance of iNTnC2 as Supplementary Note 2 on pages 20-21 in Supplementary Information, which was briefly cited in the main text (L204-206). For jGCaMP8s, the impact of reduced detectability of $[Ca^{2+}]_{resting}$ in human cells was trivial compared to its performance in *D.discoideum* cells. We thus did not include the former (jG8s in human cell), since the latter (jG8s in *D.dictyostelium* cell) would be enough informative for end users to judge the appropriate GECIs for their applications.

Accordingly, changes were made as follows:

-Description for iNTnC2 was added in the main text (L203-206) together with a citation (ref15)

“We also included recently developed jGCaMP8s as a representative of existing GECIs having high Ca^{2+} -affinity and fast kinetics⁷, since iNTnC2¹⁵, another high-affinity GECI, was found to show insufficient performance in our *in-vitro* and *in-cell* assay (**Supplementary Note 2**).”

15. Subach, O.M. et al. cNTnC and fYTnC2, Genetically Encoded Green Calcium Indicators Based on Troponin C from Fast Animals. *International journal of molecular sciences* **23**, 14614 (2022)

-Supplementary Note2 describing attenuated performance of iNTnC2 was added to the Supplementary Information (page21-22).

Supplementary Note 2

In vitro and in-cell Performance of iNTnC2

We conducted a comparative analysis to examine the possibility that iNTnC2¹⁵, another high-affinity GECI, may outperform CaMPARI-nano. N-terminally poly-histidine-tagged iNTnC2 was expressed in *E.coli* and the purified proteins were subjected to biochemical analysis. Our Ca²⁺ titration revealed the K_d of iNTnC2 to be 41 nM (**Supplementary Note fig. 1a**), a slightly smaller value than the previously reported (49 nM)¹⁵. The dynamic range ($\Delta F/F_{\min}$) of iNTnC2, originally reported to be as high as 30¹⁵, could not be replicated under our experimental conditions. While the maximum brightness of mNeonGreen-based iNTnC2 was found to be 1.15-fold higher than that of mEOS2-based CaMPARIs, its minimum brightness at Ca²⁺-saturated conditions was not low enough, leading to a significantly reduced dynamic range ($\Delta F/F_{\min} = 2.28$; **Supplementary Note fig. 1b**).

To further check whether the attenuated dynamic range persisted in human and *Dictyostelium* cells, we conducted comparative *in-cell* studies with CaMPARI-nano. In HeLa cells, we evaluated the maximum and minimum brightness of both iNTnC2 and CaMPARI-nano, both of which are inversely flashing-type GECIs. When cells were treated with 1 μ M of ionomycin in the presence of 1 mM extracellular Ca²⁺ concentration, the maximum intensity of CaMPARI-nano, normalized by co-expressed mCherry, reached 6.0 (**Supplementary Note fig. 1c**), in contrast, iNTnC2's was notably lower at 1.4. Furthermore, after treatment with 15 μ M BAPTA-AM, the minimum intensities of both CaMPARI-nano and iNTnC2 were similarly low. This led to a significantly narrower *in-cell* dynamic range for iNTnC2 ($\Delta F/F_{\min} = 5.3$) compared to CaMPARI-nano ($\Delta F/F_{\min} = 22.5$).

To further investigate the potential temperature sensitivity of *in-cell* dynamic range of iNTnC2, Ca²⁺ imaging was performed in *Dictyostelium* cells maintained at their optimal culture temperature of 23°C. We focused on fluorescent intensity under resting conditions because ionomycin and BAPTA-AM are not functional in *Dictyostelium* cells. As depicted in **Figure 2a**, both CaMPARI-nano ($K_d = 19$ nM) and CaMP2_F391W+mut3 ($K_d = 43$ nM) had similarly high resting intensities when normalized by co-expressed mRFPmars. However, despite iNTnC2's comparable K_d value and *in-vitro* brightness to CaMPARIs, its resting intensity in *Dictyostelium* cells was significantly lower than CaMPARI-nano's (**Supplementary Note fig. 2**). This indicates the potential instability of iNTnC2 in *Dictyostelium* cells, similar to observations in HeLa cells. Moreover, the [Ca²⁺]_{in} transient induced by cAMP stimulation was barely detectable with iNTnC2, suggesting the *in-cell* dynamic range of iNTnC2 was significantly lower than that of CaMPARI-nano and jGCaMP8s (**Supplementary Note fig. 2**). Importantly, physiological [Ca²⁺]_{in} transients associated with spontaneously synthesized cAMP remained undetectable with iNTnC2 (**Supplementary Note fig. 3**).

Supplementary Note fig. 1. Properties of iNTnC2. (a) Emission spectrum of purified iNTnC2. (b) Ca²⁺-titration curve. (c) *in-cell* brightness of iNTnC2. Normalized intensity of HeLa cells expressing iNTnC2_P2AmCherry or CaMPARI-nano_P2AmCherry treated with BAPTA-AM (Ca²⁺-depletion) or ionomycin (Ca²⁺-saturation). Representative 10 cells from a single experiment for each condition.

Supplementary Note fig. 2. Performance of iNTnC2 in cAMP stimulated *D.discoideum* cells. Normalized intensity change for CaMPARI-nano, iNTnC2, and jGCaMP8s. Mean \pm s.e.m.. N = 5 cells. Reproducibility was confirmed by three independent experiments. Bars indicate cAMP stimulation.

Supplementary Note fig. 3. $[\text{Ca}^{2+}]_i$ dynamics for spontaneously signaling *D.discoideum* cells. Normalized intensity of *D.discoideum* cells expressing CaMPARI-nano, iNTnC2, and jGCaMP8s. Traces of three representative ROIs each containing ~ 10 cells.